# β$_{IV}$-spectrin as a stalk cell-intrinsic regulator of VEGF signaling

Eun-A Kwak[1,12], Christopher C. Pan[2,12], Aaron Ramonett[1], Sanjay Kumar ® [3], Paola Cruz-Flores[4], Tasmia Ahmed[4], Hannah R. Ortiz[1], Jeffrey J. Lochhead[1], Nathan A. Ellis[5], Ghassan Mouneimne[5], Teodora G. Georgieva[6], Yeon Sun Lee[1], Todd W. Vanderah[1], Tally Largent-Milnes[1], Peter J. Mohler[7], Thomas J. Hund[8], Paul R. Langlais[9], Karthikeyan Mythreye[10] & Nam Y. Lee ® [1,4,11✉]

Defective angiogenesis underlies over 50 malignant, ischemic and inflammatory disorders yet long-term therapeutic applications inevitably fail, thus highlighting the need for greater understanding of the vast crosstalk and compensatory mechanisms. Based on proteomic profiling of angiogenic endothelial components, here we report β$_{IV}$-spectrin, a non-erythrocytic cytoskeletal protein, as a critical regulator of sprouting angiogenesis. Early loss of endothelial-specific β$_{IV}$-spectrin promotes embryonic lethality in mice due to hypervascularization and hemorrhagic defects whereas neonatal depletion yields higher vascular density and tip cell populations in developing retina. During sprouting, β$_{IV}$-spectrin expresses in stalk cells to inhibit their tip cell potential by enhancing VEGFR2 turnover in a manner independent of most cell-fate determining mechanisms. Rather, β$_{IV}$-spectrin recruits CaMKII to the plasma membrane to directly phosphorylate VEGFR2 at Ser984, a previously undefined phosphoregulatory site that strongly induces VEGFR2 internalization and degradation. These findings support a distinct spectrin-based mechanism of tip-stalk cell specification during vascular development.

[1] Department of Pharmacology, University of Arizona, Tucson, AZ 85724, USA. [2] Department of Pharmacology and Cancer Biology, Duke University, Durham, NC 27710, USA. [3] Division of Biology, Indian Institute of Science Education and Research, Tirupati 517507, India. [4] Department of Chemistry & Biochemistry, University of Arizona, Tucson, AZ 85724, USA. [5] Cellular and Molecular Medicine, University of Arizona, Tucson, AZ 85724, USA. [6] BIO5 Institute, University of Arizona, Tucson, AZ 85724, USA. [7] Department of Physiology and Cell Biology, Ohio State University, Columbus, OH 43210, USA. [8] Department of Biomedical Engineering, Ohio State University, Columbus, OH 43210, USA. [9] Department of Internal Medicine, University of Arizona, Tucson, AZ 85724, USA. [10] Department of Pathology, University of Alabama at Birmingham, Birmingham, AL 35294, USA. [11] Cancer Center, University of Arizona, Tucson, AZ 85724, USA. [12] These authors contributed equally: Eun-A Kwak, Christopher C. Pan. ✉email: namlee@email.arizona.edu

Angiogenesis is the formation of new blood vessels from preexisting vasculature, an essential process during development but also underlies the pathogenesis of over 50 malignant, ischemic and inflammatory diseases[1,2]. Bevacizumab, ranibizumab, and sorafenib are among the most widely used vascular endothelial growth factor (VEGF) signaling antagonists that inhibit tumor vasculatures and abnormal vessel growth in patients with macular degeneration or proliferative diabetic retinopathy[3–5]. But while initially effective, nearly all existing angiogenic inhibitors fail to provide long-term benefits due to acquired resistance and various complications[6–8], thus highlighting the growing need to identify compensatory mechanisms of VEGF receptor (VEGFR) signaling and crosstalk pathways as vascular targets to realize the full potential of antiangiogenic therapies.

Among a family of VEGFRs (VEGFR1-3), VEGFR2 is considered the primary driver of angiogenesis since its activation, internalization and endocytic trafficking regulate the specificity, duration, and amplitude of many, if not most, of the VEGF-induced signaling pathways[9–11]. But unlike the early events surrounding VEGFR2 activation and endocytic recycling, how this receptor is deactivated and degraded remains much less clear. Indeed, while many important tyrosine phosphatases have been identified to target VEGFR2, their selectivity for particular tyrosine residues is yet to be defined[12–17]. Moreover, the role of serine/threonine phosphorylation in receptor turnover remains quite elusive. Protein kinase C (PKC) is among the few reported instances where such a kinase has been shown to promote VEGFR2 lysosomal degradation, although the precise phosphoregulatory site(s) has not been mapped, nor is it clear whether this kinase directly or indirectly phosphorylates the receptor[18].

But an in-depth molecular understanding of VEGFR2 deactivation is particularly important considering that endothelial tip cells require some of the highest levels of VEGF/VEGFR2 signaling and thus serve as attractive therapeutic targets[9,19–21]. These highly specialized tip endothelial cells (EC) consequently increase Dll4 expression, which in turn, activate Notch signaling to promote stalk cell behavior in adjoining ECs[22]. Although VEGFR1, VEGFR3, and Nrp1 represent additional modulators of Notch/Jagged1 signaling to determine EC fate, the local fluctuations in VEGFR2 levels arguably describe the most critical and dynamically competitive process of tip/stalk cell positioning[21,23,24]. Hence, precisely how one EC is able to upregulate VEGFR2 expression in the first place over neighboring ECs to gain the initial tip-cell advantage remains a fundamental question.

Spectrins are cytoskeletal scaffolding proteins required for cell membrane structural organization and maintenance of integral proteins including ion channels and transporters[25]. There are two α- and five β-subunits in the mammalian system that form heterotetrameric complexes[26]. While most spectrins are widely expressed, $\beta_{IV}$-spectrin is a non-erythrocytic member with a highly restricted expression pattern, so far characterized only in the nervous system, heart, and pancreatic β cells[27–29]. $\beta_{IV}$-spectrin interacts with other cytoskeletal and scaffolding proteins including actin, ankyrins, and membrane channels, and its functional mutations are associated with congenital myopathy and neuropathy[30].

Here we show $\beta_{IV}$-spectrin as an endothelial cell (EC) component required for normal sprouting angiogenesis. We find that this membrane-associated scaffold protein plays a critical role in regulating VEGF signaling by inducing VEGFR2 turnover to maintain tip/stalk cell balance during vascular sprouting.

## Results

### Identification of $\beta_{IV}$-spectrin expression in vascular ECs.
We performed quantitative proteomics on mouse embryonic endothelial cell (MEEC) lysates harvested at either 50% or 100% cell density to identify proteins specifically associated with angiogenic log-phase growth but not confluent, quiescent state (Figure S1A). Preliminary results confirmed a number of established angiogenic markers but also identified several unanticipated candidates that have never been shown to be expressed in ECs including $\beta_{IV}$-spectrin—a membrane-associated cytoskeletal protein characterized in heart and neuronal tissues but not in the vascular system (Figure S1B). Consistent with this finding, subsequent western analysis showed that $\beta_{IV}$-spectrin expresses robustly in MEECs particularly during log-phase growth but not when cells reach high confluence (Fig. 1A; graph quantification). $\beta_{IV}$-spectrin was also expressed at varying levels in many different EC types including those of microvascular, venous, and aortic origin, which we were able to silence via shRNA-mediated knockdown (Fig. 1B). To begin exploring its function, we selected MEECs as the primary model system for in vitro angiogenesis assays and observed that $\beta_{IV}$-spectrin knockdown ($\beta_{IV}$-shRNA) significantly enhances cell migration and proliferation while also promoting endothelial capillary branching more robustly than control wildtype (WT) cells (Fig. 1C–E). Taken together, these preliminary functional studies suggested that $\beta_{IV}$-spectrin acts as a negative regulator of angiogenesis.

### $\beta_{IV}$-spectrin regulates sprouting angiogenesis in vivo.
To test whether $\beta_{IV}$-spectrin is indeed a critical component of angiogenesis and/or endothelial functions in vivo, an inducible EC-specific knockout mouse model was generated by crossing floxed $\beta_{IV}$-spectrin mice (Spnb4$^{fl/fl}$)[31] with an EC-selective Cdh5(PAC)-CreERT2 model[32] that facilitates tamoxifen-induced $\beta_{IV}$-spectrin knockout ($\beta_{IV}$-EC$^{KO}$). Here, early knockout starting at E8.5 with tamoxifen injection resulted in partial lethality, with approximately one-third to half of the embryos dying around E11.5–12.5 (Fig. 2A). Non-resorbed $\beta_{IV}$-EC$^{KO}$ embryos at these stages displayed signs of hemorrhaging in various sites of the head and trunk and frequent bulging protrusions in the forebrain and hindbrain regions (Fig. 2A; arrows). Immunofluorescent staining with CD31, an EC marker, showed a marked increase in the number of major cranial vessels along with hypersprouting and branching at the distal ends of the vessels in $\beta_{IV}$-EC$^{KO}$ embryos (Fig. 2B). In a separate experiment, $\beta_{IV}$-EC$^{KO}$ embryos harvested at E15.5 displayed similar signs of bleeding and were generally smaller than WT but otherwise appeared normal and survived to term (Figure S2).

Given these variations in embryonic viability, we focused on the retinal vasculature as a model system to specifically examine how $\beta_{IV}$-spectrin controls sprouting angiogenesis. Here, developing retinas were collected from P5 pups upon tamoxifen injections at P1 and P3 to avoid potential embryonic lethality (Fig. 2C). While isolectin B4 (IB4) staining showed that the overall radial vascular expansion was similar between the two groups, $\beta_{IV}$-EC$^{KO}$ retinas displayed hyperbranching and greater vascular density than WT (Fig. 2C; inset a–d and graph). Staining with ERG, an EC-specific nuclear transcription factor, also showed increased density in the P5 $\beta_{IV}$-EC$^{KO}$ retina, suggesting that $\beta_{IV}$-spectrin deficiency may promote EC hyperproliferation (Fig. 2D). Interestingly, $\beta_{IV}$-spectrin expression was predominantly observed in the periphery of the developing vascular plexus of WT retina when compared to $\beta_{IV}$-EC$^{KO}$, which showed strong tamoxifen-induced depletion both in retina and isolated primary ECs, suggesting that this cytoskeletal protein is most active where new endothelial sprouts are forming (Fig. 2E, F). To further test whether $\beta_{IV}$-spectrin expression is mainly localized to the endothelial lining of the vessels and is absent from the lumen, confocal Z-stacking images were obtained at high magnification near the leading edge of WT P5 retina

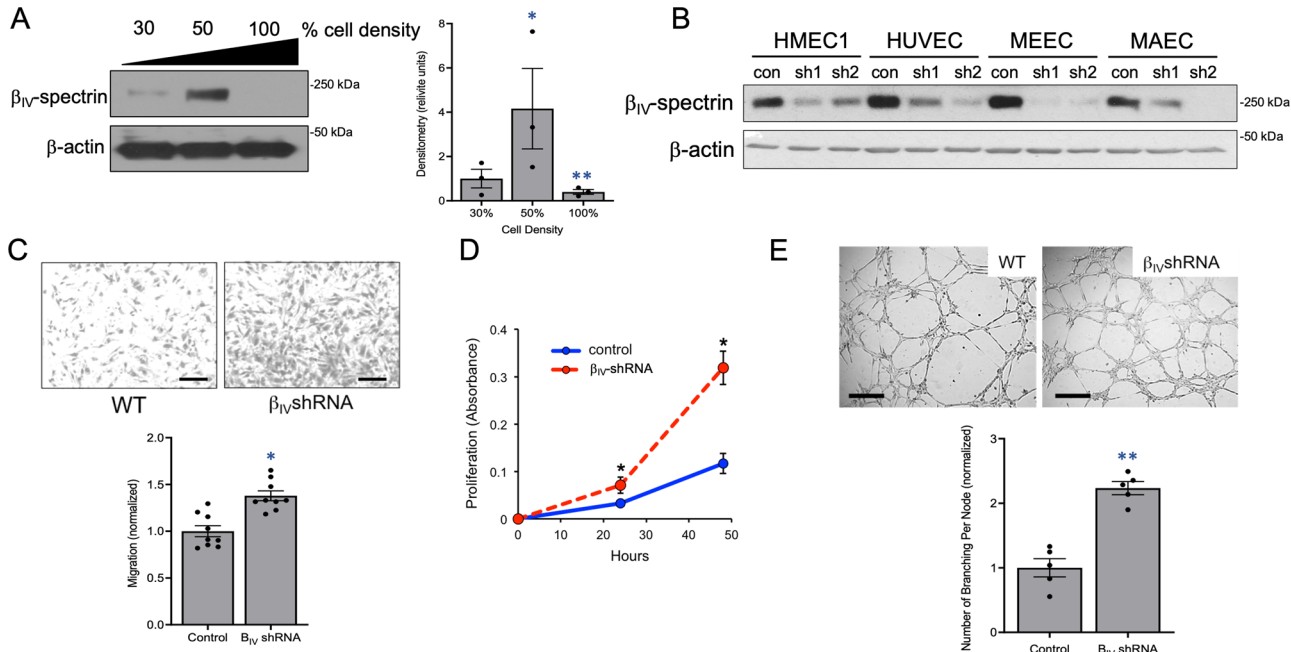

**Fig. 1 $\beta_{IV}$-spectrin expression in ECs and its regulation of vascular sprouting. A** Western blot and densitometry quantification show endogenous $\beta_{IV}$-spectrin expression in MEECs at the designated cell densities, where $n = 3$ independent experiments. Each densitometry value was normalized by the mean value of the 30% mean and then expressed as a fold change over the 30% value. Error bars represent SEM and type 2 $t$-test results show: *$p = 0.04$, **$p = 0.02$ relative to 30% cell density. Source data are provided as a Source data file. **B** Western blot shows endogenous $\beta_{IV}$-spectrin expression and its knockdown using two distinct shRNA target sequences in human microvascular EC 1 (HMEC1) and human umbilical vein EC (HUVEC) along with MEEC and primary mouse aortic ECs (MAEC) ECs. **C** Representative images of transwell migration using WT (control scrambled) and stable knockdown $\beta_{IV}$-shRNA MEECs where $n = 9$ regions of interest from 3 independent experiments. Scale bar: 100 µm. Data are presented as normalized to the WT ± SEM. Type 2 $t$-test results show relative to control: *$p = 0.00001$. Source data are provided as a Source data file. **D** Crystal violet assay shows comparison of proliferation between WT and $\beta_{IV}$-shRNA MEECs over 48 h. Graph represents normalized average from three independent experiments. Error bars represent SEM and type 2 $t$-test results show relative to control: *$p = 2.77E^{-9}$. Source data are provided as a Source data file. **E** Matrigel-induced angiogenesis assay shows EC differentiation and branching in WT versus $\beta_{IV}$-shRNA MEECs 20 h upon plating. Scale bar: 100 µm. Data representative of $n = 5$ regions of interest from 3 independent experiments presented as normalized values to the control. Error bars represent SEM and type 2 $t$-test results show: *$p = 0.000081$ relative to control. Source data are provided as a Source data file.

(Fig. 2G). Three-dimensional cross-sections of IB4 (red) and $\beta_{IV}$-spectrin (green) revealed their prominent staining along the circumference of the capillary vessels that at least partially co-localized (Fig. 2G; cross-section regions indicated by arrows), hence suggesting that $\beta_{IV}$-spectrin staining is mostly observed along the endothelium of sprouting retinal vessels.

**$\beta_{IV}$-spectrin-dependent changes in VEGFR2 protein expression.** To identify mechanisms by which $\beta_{IV}$-spectrin regulates sprouting angiogenesis, we performed comparative quantitative proteomics on WT and $\beta_{IV}$-shRNA ECs and found over 600 differentially regulated proteins—240 of which were significantly downregulated in $\beta_{IV}$-spectrin depleted cells whereas, conversely, nearly 400 proteins were statistically upregulated (Fig. 3A; Figure S3). Among the top statistically significant responders, VEGFR2 was found increased three-fold upon $\beta_{IV}$-spectrin knockdown, a finding that was subsequently confirmed via western analysis of the total receptor expression and its autophosphorylation levels compared to WT (Fig. 3B). Consistent with the proteomics analysis, many VEGFR2 downstream pathways were also enhanced in $\beta_{IV}$-shRNA and $\beta_{IV}$-EC$^{KO}$-derived ECs including Akt, p38, PLC, and eNOS (Fig. 3C, D). However, ERK activation was impaired in both $\beta_{IV}$-shRNA and $\beta_{IV}$-EC$^{KO}$ cells, a finding that was consistent with previous reports that the early endocytic trafficking of VEGFR2 is required for efficient ERK activation[16].

The increased VEGFR2 expression associated with $\beta_{IV}$-spectrin depletion was not due to gene upregulation, as RT-qPCR analysis showed its transcript level to be actually lower than control cells, conceivably due to a negative feedback response (Fig. 3E). Instead, $\beta_{IV}$-spectrin appeared to promote receptor turnover since blocking the lysosomal pathway for a brief period (2–4 h) dramatically elevated its protein level similar to that of $\beta_{IV}$-spectrin depleted cells (Fig. 3F). Hence, to test whether $\beta_{IV}$-spectrin is directly linked to VEGFR2 turnover, we reconstituted $\beta_{IV}$-spectrin in the mouse-derived $\beta_{IV}$-shRNA cells with a human expression construct and found that ectopic overexpression strongly reduced VEGFR2 levels in both WT and $\beta_{IV}$-shRNA ECs (Fig. 3G). Similar outcomes were observed in vivo where VEGFR2 levels were dramatically increased (Fig. 3H; graph), particularly along the leading edge of vascular expansion in $\beta_{IV}$-EC$^{KO}$ retina compared to WT (Fig. 3I; arrows). Again, high magnification confocal Z-stack images of capillaries near the leading edge showed prominent VEGFR2 (green) and IB4 (red) staining along the endothelial lining but were largely excluded from the lumen, suggesting that the staining of VEGFR2 near the radial front of vascular expansion were specific (Fig. 3J). Moreover, co-staining of VEGFR2 and ERG revealed a greater concentration of ECs near the radial front in $\beta_{IV}$-EC$^{KO}$ retina (Figure S4A), indicating that these $\beta_{IV}$-spectrin-dependent changes occur predominantly at the active sites of sprouting angiogenesis. Accordingly, we detected weak $\beta_{IV}$-spectrin expression in the fully developed

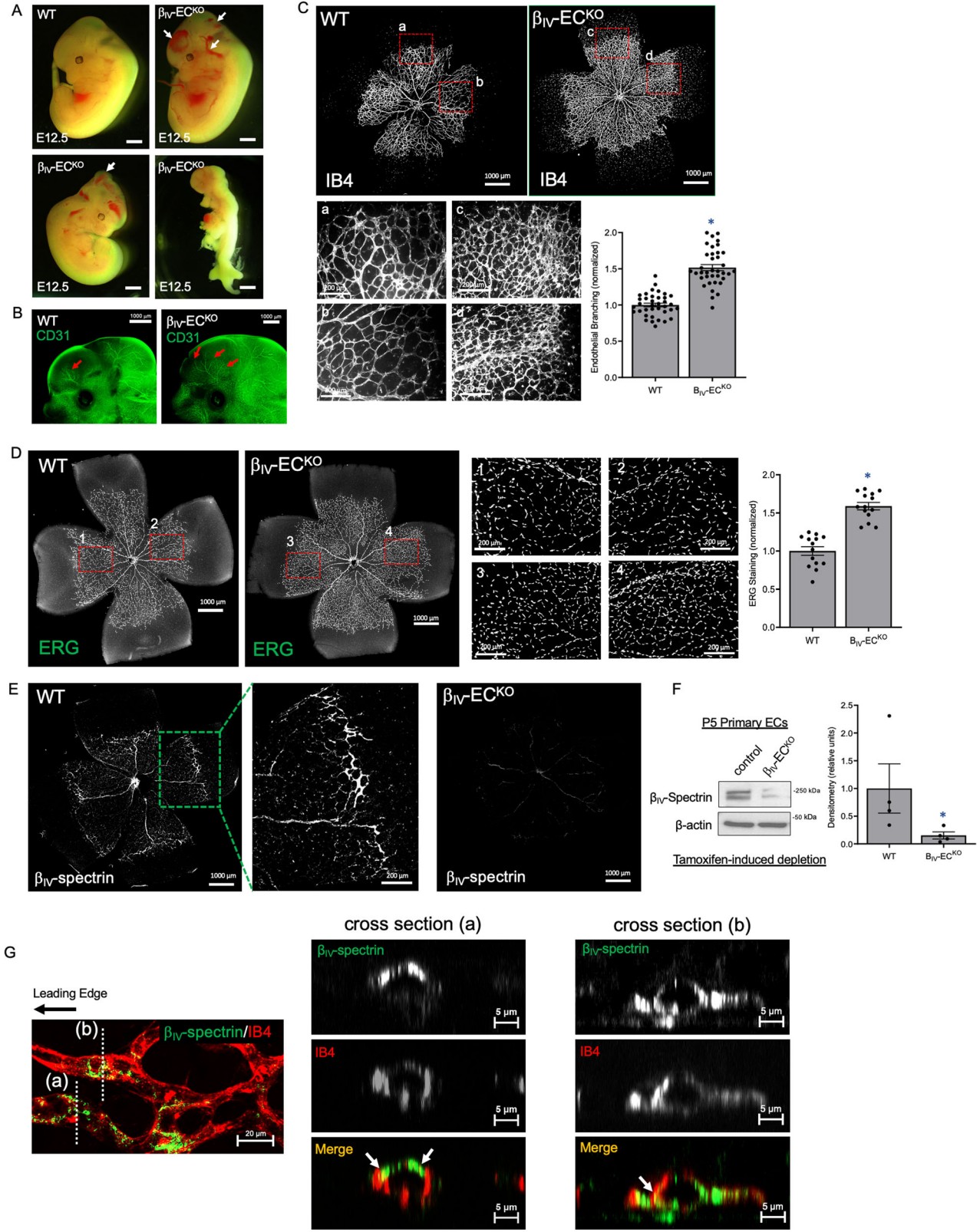

vasculature of adult mouse retina (Figure S4B). But surprisingly, the VEGFR2 expression was still moderately higher in mature vessels of $\beta_{IV}$-EC$^{KO}$ retina relative to WT, suggesting that even minimal $\beta_{IV}$-spectrin expression in developed vasculature may be sufficient to limit VEGFR2 levels and maintain quiescence (Figure S4C).

**$\beta_{IV}$-spectrin/CaMKII complex promotes VEGFR2 degradation.** To understand how $\beta_{IV}$-spectrin controls angiogenesis at the cellular level, we began by assessing its subcellular localization. Here, confocal Z-stack analysis of endogenous $\beta_{IV}$-spectrin revealed its distinct punctate distribution along the basolateral membrane compartments when compared to counterstaining

**Fig. 2 EC-specific $\beta_{IV}$-spectrin deletion in mice results in partial embryonic lethality and defective sprouting angiogenesis. A** Representative images of control and $\beta_{IV}$-EC$^{KO}$ mouse embryos at E12.5 upon tamoxifen injection at E8.5. Compared to WT (control-Cdh5-creERT2), $\beta_{IV}$-EC$^{KO}$ embryos display hemorrhaging, bulging hindbrain (white arrows), or resorption. Scale bar: 1000 μm. **B** Fluorescence images show tamoxifen-treated WT and $\beta_{IV}$-EC$^{KO}$ embryos harvested at E12.5 and stained for CD31 (green). Significantly greater vessel sprouting is observed in $\beta_{IV}$-EC$^{KO}$ (red arrows). **C** Images show IB4 staining (red) of WT and $\beta_{IV}$-EC$^{KO}$ neonatal retina (P5) upon tamoxifen treatment at P1 and P3. Inset images (a–d) show vessel branching upon $\beta_{IV}$-spectrin depletion. Data are presented as mean values ± SEM normalized to the control. Type 2 $t$-test results show: *$p = 1.3E^{-15}$ relative to WT ($n = 37$ of at least 6 mice retina per group). Source data are provided as a Source data file. **D** Images show ERG staining (green) of WT and $\beta_{IV}$-EC$^{KO}$ neonatal retina (P5) upon tamoxifen treatment at P1 and P3. Inset (1–4) and graph demonstrate EC-specific counts. Data representative of $n = 14$ measurements from 3 separate mouse retina per group represented as a mean value ± SEM and normalized to the WT. A Type 2 $t$-test results show: *$p = 1.01E^{-8}$ relative to WT. Source data are provided as a Source data file. **E** Representative image of $\beta_{IV}$-spectrin staining (red) in WT and $\beta_{IV}$-EC$^{KO}$ neonatal retina (P5). Inset image shows $\beta_{IV}$-spectrin concentrated along the periphery of the vascular plexus in WT whereas staining is significantly diminished in $\beta_{IV}$-EC$^{KO}$. Data representative of 8 retina per group. **F** Biochemical analysis shows nearly 90% depletion of endogenous $\beta_{IV}$-spectrin in primary ECs isolated from control and $\beta_{IV}$-EC$^{KO}$ mice at P5. Data are presented as mean values ± SEM of 4 independent analyses normalized to the WT values. Type 2 $t$-test results show *$p = 0.05$ over WT. Source data are provided as a Source data file. **G** Confocal Z-stack images acquired with ×63 oil objective show cross-sections of the WT P5 mouse retinal capillaries near the leading edge of vascular expansion co-stained for $\beta_{IV}$-spectrin (green) and IB4 (red). White arrows indicate the patches of co-localization (yellow).

with a known endocytic vesicle marker, EEA1, which displayed prominent cytoplasmic localization (Fig. 4A). And consistent with our biochemical findings, immunofluorescence co-staining for endogenous $\beta_{IV}$-spectrin and VEGFR2 showed the expected increase in VEGFR2 levels in $\beta_{IV}$-shRNA cells, although the Z-stack images showed a far greater accumulation of this receptor along the basolateral compartments than control, strongly suggesting that $\beta_{IV}$-spectrin somehow mediates the internalization and degradation of VEGFR2 (Figure S5, arrows).

In seeking out the molecular basis for these effects on VEGFR2, previous studies have defined a key role for $Ca^{2+}$/calmodulin-dependent protein kinase II (CaMKII) in binding to the C-terminal region of $\beta_{IV}$-spectrin, which causes the phosphorylation-dependent regulation of voltage-gated sodium channels in cardiomyocyte excitability[29]. Indeed, mutant $\beta_{IV}$-spectrin mouse models such as $\beta_{IV}$-qv$^{4J}$, which harbors a C-terminal deletion including the peptide segment comprising the CaMKII-binding site (Fig. 4B), fails to recruit CaMKII to the membrane, resulting in cardiac dysfunction. Hence, we tested whether CaMKII is involved in $\beta_{IV}$-spectrin-dependent VEGFR2 turnover first by overexpressing this kinase. Notably, we discovered that ectopic CaMKII expression strongly reduced VEGFR2 levels in control but not $\beta_{IV}$-shRNA ECs, suggesting that CaMKII requires the interaction with $\beta_{IV}$-spectrin to promote VEGFR2 degradation (Fig. 4C; graph). In a reciprocal experiment, transiently blocking CaMKII activity with a small molecule inhibitor, KN-93, resulted in increased VEGFR2 expression in WT (Fig. 4D). In fact, while KN-93 treatment caused VEGFR2 levels to reach levels similar to or even slightly higher than that of chloroquine, a lysosome inhibitor, no discernable changes in VEGFR2 levels were observed in $\beta_{IV}$-shRNA ECs, which remained constitutively high regardless of KN-93 or chloroquine treatment (Fig. 4D; graph).

**CaMKII-induced VEGFR2 phosphorylation and degradation**. Based on the above results, we hypothesized that $\beta_{IV}$-spectrin forms a signaling complex with CaMKII along the plasma membrane to cause phosphorylation-induced VEGFR2 turnover. To test this possibility, we first evaluated their novel interaction via endogenous $\beta_{IV}$-spectrin immunoprecipitation and detected its association with CaMKII and VEGFR2 at basal state (Fig. 4E). The formation of this trimeric complex was further supported by our immunofluorescence studies in intact cells where endogenous $\beta_{IV}$-spectrin strongly co-localized with both CaMKII and VEGFR2 in control cells but was nearly abolished upon $\beta_{IV}$-spectrin depletion (Fig. 4F; white arrows indicating trimeric co-localization and graph quantification). Next, to test how VEGF-induced receptor activation affects this interaction, a time-course

was performed in which cells were stimulated for the indicated time points prior to $\beta_{IV}$-spectrin/CaMKII immunoprecipitation. Relative to basal state, there was a sharp increase in VEGFR2 interaction the 0 to 30 min interval before gradually subsiding, whereas CaMKII constitutively interacted with $\beta_{IV}$-spectrin at basal state but gradually dissociated over time in a VEGF-dependent manner (Fig. 4G; upper and lower graphs), suggesting that the preformed $\beta_{IV}$-spectrin/CaMKII complex primarily targets the receptor once activated.

To determine whether the $\beta_{IV}$-spectrin/CaMKII complex induces VEGFR2 proteolysis by promoting receptor internalization, we measured cell surface levels of endogenous VEGFR2 in non-permeabilized primary ECs isolated from $\beta_{IV}$-qv$^{4J}$ mutant mice, which as mentioned earlier, lack the C-terminal CaMKII-binding site. Here, the VEGFR2 membrane retention increased by two to three-fold upon CaMKII inhibition in control ECs, whereas $\beta_{IV}$-qv$^{4J}$ cells displayed constitutively high receptor levels independent of the CaMKII activity (Fig. 5A; graph). In a parallel experiment, VEGF stimulation caused modest fluctuations in control cells, likely from the cumulative effect of receptor endocytosis, recycling and exocytosis involving rapid mobilization to the plasma membrane from the Golgi apparatus (Figure S6A)[33]. However, there was a steady rise in cell surface VEGFR2 upon VEGF stimulation in $\beta_{IV}$-qv$^{4J}$ ECs, a pattern that was similarly observed in $\beta_{IV}$-shRNA cells compared to control MEECs (Figure S6A, B), suggesting that the $\beta_{IV}$-spectrin/CaMKII complex at the membrane mediates VEGF-induced internalization and lysosomal trafficking. Indeed, we discovered that VEGFR2 ubiquitination was higher in WT compared to $\beta_{IV}$-shRNA ECs but markedly reduced, along with increased receptor expression, when CaMKII activity was blocked (Fig. 5B).

We next tested this CaMKII-dependent mechanism in vivo by comparing the VEGFR2 levels in the presence of CaMKII inhibition during retinal development. Here, P5 retinal harvest after an intraperitoneal administration of KN-93 at P1 and P3 showed a dramatic increase in VEGFR2 expression over vehicle-treated mice especially along the periphery of the vascular plexus, whereas minimal change was observed in $\beta_{IV}$-EC$^{KO}$ mice (Fig. 5C; graph), thus indicating that the spatially dynamic expression of the $\beta_{IV}$-spectrin/CaMKII complex critically regulates VEGFR2 turnover and downstream signaling and turnover primarily at the actively sprouting sites. Supporting these conclusions, we further discovered that $\beta_{IV}$qv$^{4J}$ mice have remarkably similar vascular defects due to the loss of CaMKII binding, as their retina displayed greater vascular sprouting and branching density while their isolated primary ECs showed a more diffuse $\beta_{IV}$-spectrin distribution than control along with

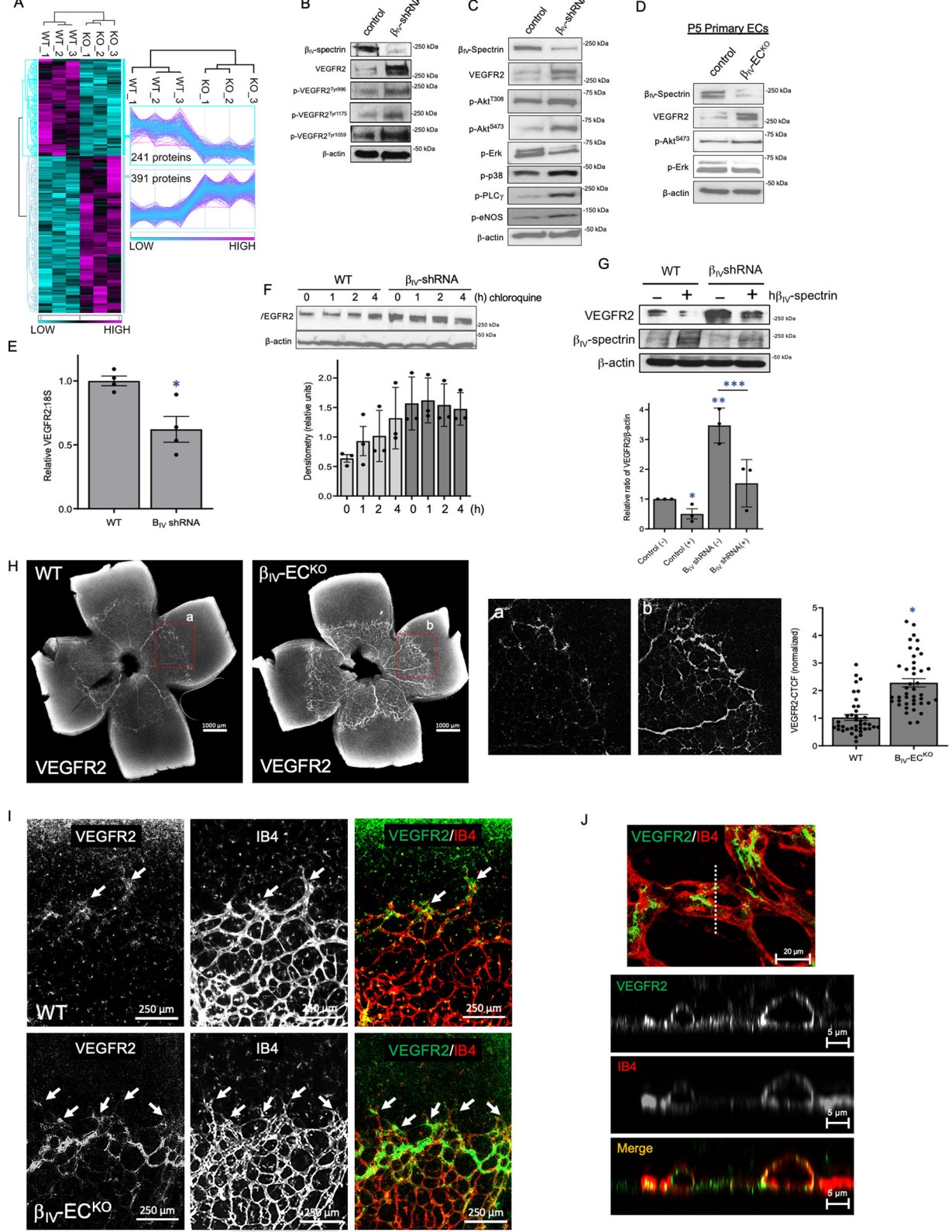

markedly higher overall VEGFR2 expression and activation of the major downstream signaling pathways (Figure S7A, B).

In testing whether CaMKII directly phosphorylates VEGFR2, we initially found at least 6 distinct sites in the receptor protein sequence that contains the CaMKII consensus phosphorylation sites R/K-XX-S/T. To identify which of these recognition sites are targeted by CaMKII in a $\beta_{IV}$-spectrin-dependent manner, we performed phosphoproteomic analysis on control and $\beta_{IV}$-EC$^{KO}$ ECs. Indeed, here we discovered two prominent sites on VEGFR2, serine984 and serine1235, which were part of the consensus CaMKII substrate motif and associated with $\beta_{IV}$-spectrin expression (Figure S8). Accordingly, we mutated these individual sites to

**Fig. 3 $\beta_{IV}$-spectrin regulates VEGFR2 expression and downstream signaling. A** Heat map based on MS-based quantitative proteomics shows differential expression of proteins found in WT versus $\beta_{IV}$-shRNA MEECs. $N = 3$ biological replicates per group. **B** Representative western blots show comparison of total VEGFR2 and activated receptor levels in WT and $\beta_{IV}$-shRNA MEECs based on pan-antibody and phosphor-Tyr-specific sites as indicated. Data representative of three independent experiments. **C**, **D** Representative western analysis of VEGFR2 and downstream signaling in control versus $\beta_{IV}$-shRNA MEECs or primary $\beta_{IV}$-EC$^{KO}$ ECs. Data representative of three independent experiments. **E** Graph shows relative VEGFR2 mRNA levels in control vs $\beta_{IV}$-shRNA MEECs, where $n = 3$ independent experiments. A Type 2 $t$-test results show relative to WT: *$p = 0.01$. Source data are provided as a Source data file. **F** Representative western and densitometry quantification graph demonstrate VEGFR2 levels at indicated time points upon chloroquine treatment (50 μM). Data are presented as mean values ± SEM of three independent experiments. A Type 2 $t$-test results show: *$p$ is 0.045 or lower relative to 0 h time point in WT. Source data are provided as a Source data file. **G** Representative western images of VEGFR2 and $\beta_{IV}$-spectrin levels upon ectopic expression of human $\beta_{IV}$-spectrin construct in WT and $\beta_{IV}$-shRNA MEECs. Graph shows normalized densitometry quantification of three independent experiments as mean values ± SEM. A Type 2 $t$-test results show *$p = 0.049$, **$p = 0.001$, ***$p = 0.015$ compared to WT or as indicated. Source data are provided as a Source data file. **H** Immunofluorescence images show VEGFR2 levels in P5 retina of WT and $\beta_{IV}$-EC$^{KO}$ mice, where $n = 39$ regions of interest for CTCF in 3 separate mouse retina per group in the WT and $n = 42$ regions of interest for CTCF in 3 separate mouse retina per group in $\beta_{IV}$-EC$^{KO}$. Graph shows relative fluorescence of VEGFR2 as quantified by CTCF (corrected total cell fluorescence). A Type 2 $t$-test results show: *$p = 3.87E^{-14}$ over WT. Data are presented as mean values ± SEM. Source data are provided as a Source data file. **I** Immunofluorescence co-staining of VEGFR2 (green) and IB4 (red) demonstrates concentrated receptor expression along the radial front of vascular expansion (white arrows) in $\beta_{IV}$-EC$^{KO}$ retina. Data representative of $n = 8$ retina per group. Source data are provided as a Source data file. **J** Confocal Z-stack images acquired with 63X oil objective show cross-sections of the WT P5 mouse retinal capillaries near the leading edge of vascular expansion co-stained for VEGFR2 (green) and IB4 (red). White arrows indicate the patches of co-localization (yellow).

alanine (S984A and S1235A) to determine whether can resist CaMKII-dependent VEGFR2 degradation (Fig. 6A, B). Here, the ectopic expression of WT, S984A, and S1235A all yielded similar levels whereas co-overexpression with CaMKII resulted in a dramatic reduction of both WT and S1235A, thus indicating that S1235 does not mediate CaMKII-induced VEGFR2 degradation (Fig. 6B). However, expression of the S984A mutant yielded constitutively high expression irrespective of CaMKII, thus indicating that this is a major phosphoregulatory site directly targeted by CaMKII to induce VEGFR2 lysosomal degradation (Fig. 6B). Consistent with this notion, their expression in nonpermeabilized cells showed that co-expression with CaMKII reduced the cell surface distribution of WT and S1235A by more than 50% whereas S984A levels remained constant, thus supporting a direct role for the $\beta_{IV}$-spectrin/CaMKII complex in S984 phosphorylation-induced VEGFR2 internalization towards lysosomal degradation (Fig. 6C).

**$\beta_{IV}$-spectrin controls tip/stalk cell dynamics.** Because we found $\beta_{IV}$-spectrin expression to be primarily concentrated near the leading edge of sprouting vessels to regulate VEGFR2 expression, we reasoned that this cytoskeletal protein acts as an inhibitor of tip-cell behavior through the Dll4/Notch pathway. Indeed, our preliminary biochemical analysis was consistent with this hypothesis since tip-cell markers such as CD34 and Dll4 were preferentially expressed in $\beta_{IV}$-shRNA ECs whereas prominent stalk cell markers such as Notch1 and Jagged1 were down-regulated (Fig. 7A; graphs). However, other prominent stalk cell phenotypes such as lower VEGFR1 levels were not observed upon $\beta_{IV}$-spectrin depletion, and in fact, was increased (Fig. 7A; graphs), suggesting that $\beta_{IV}$-spectrin regulation of tip/stalk cell specification is not Notch-dependent. Although this finding argued for a more direct role for $\beta_{IV}$-spectrin in tip-stalk cell selection through regulation of VEGFR2 protein turnover, we also explored the TGF-β and BMP9/10 signaling as an alternative mechanism. Since previous studies have established a pivotal role for Smad signaling in promoting stalk cell competence, we tested whether $\beta_{IV}$-spectrin controls the activation of downstream Smad1/5 and Smad2/3 transcriptional mediators that are known to promote the expression of stalk cell-associated genes[34,35]. Biochemical analysis showed that both BMP9 and TGF-β efficiently activated Smad1/5 and Smad2/3 effectors, respectively, independent of $\beta_{IV}$-spectrin expression (Figure S10A). Similarly,

loss of $\beta_{IV}$-spectrin did not alter the expression of the BMP9 and TGF-β-responsive receptors including ALK1, ALK5, and TGFβRII (Figure S10B), indicating that $\beta_{IV}$-spectrin does not regulate VEGFR2 levels or tip-stalk cell dynamics through TGF-β signaling. Still, the notion that $\beta_{IV}$-spectrin strongly promotes stalk cell identity was further evidenced by the fact that CD34-positive cells were present in far greater numbers in $\beta_{IV}$-shRNA than control (Fig. 7B). And while VEGF stimulation did not increase the overall CD34+ cell counts, $\beta_{IV}$-spectrin depletion enhanced the ligand-responsiveness toward the formation of filopodial protrusions which reflected the hallmark characteristics of tip cells (Fig. 7C).

Lastly, the role of $\beta_{IV}$-spectrin in tip-stalk cell potential was further confirmed in vivo as we observed a sharp increase in CD34 levels in P5 retinal vasculature of $\beta_{IV}$-EC$^{KO}$ mice along with greater number of tip cells with filopodial projections along the sprouting radial front (Fig. 7D, E). More importantly, $\beta_{IV}$-spectrin/CD34 co-staining revealed that $\beta_{IV}$-spectrin expression was actively excluded from tip cells and more confined to a narrow band of stalk cells near the vascular front to demonstrate its spatially dynamic role in stalk cell maintenance, a result that was similarly observed using another tip-cell marker, ESM1 (Fig. 7F, Figure S9A–C). This tip-cell suppression, by way of promoting the stalk cell phenotype, was also evident during embryonic development where higher counts of CD34 puncta were apparent in the cranial vasculature of $\beta_{IV}$-EC$^{KO}$ compared to control embryos (Fig. 7G). Together, these findings support $\beta_{IV}$-spectrin as a stalk cell-specific regulator of VEGFR2 activity required for proper tip/stalk cell specification during sprouting angiogenesis and vascular remodeling.

## Discussion
Despite recent advances, much of our current understanding of endothelial tip and stalk cell specification relates to events downstream of VEGFR2, wherein tip cells increase VEGF-induced Dll4 expression which, in turn, activates Notch signaling in stalk cells to suppress the tip-cell phenotype. Over the years many essential mediators of the Notch pathway have been characterized, including Wnt, SIRT1, and Jagged—all of which help maintain their respective tip or stalk cell behavior[36]. Yet there has been limited progress in deciphering the early stages of nascent sprouting when ECs begin to compete for increased VEGFR2 expression. While it is believed that the local VEGFR2 levels dynamically fluctuate until one cell gains

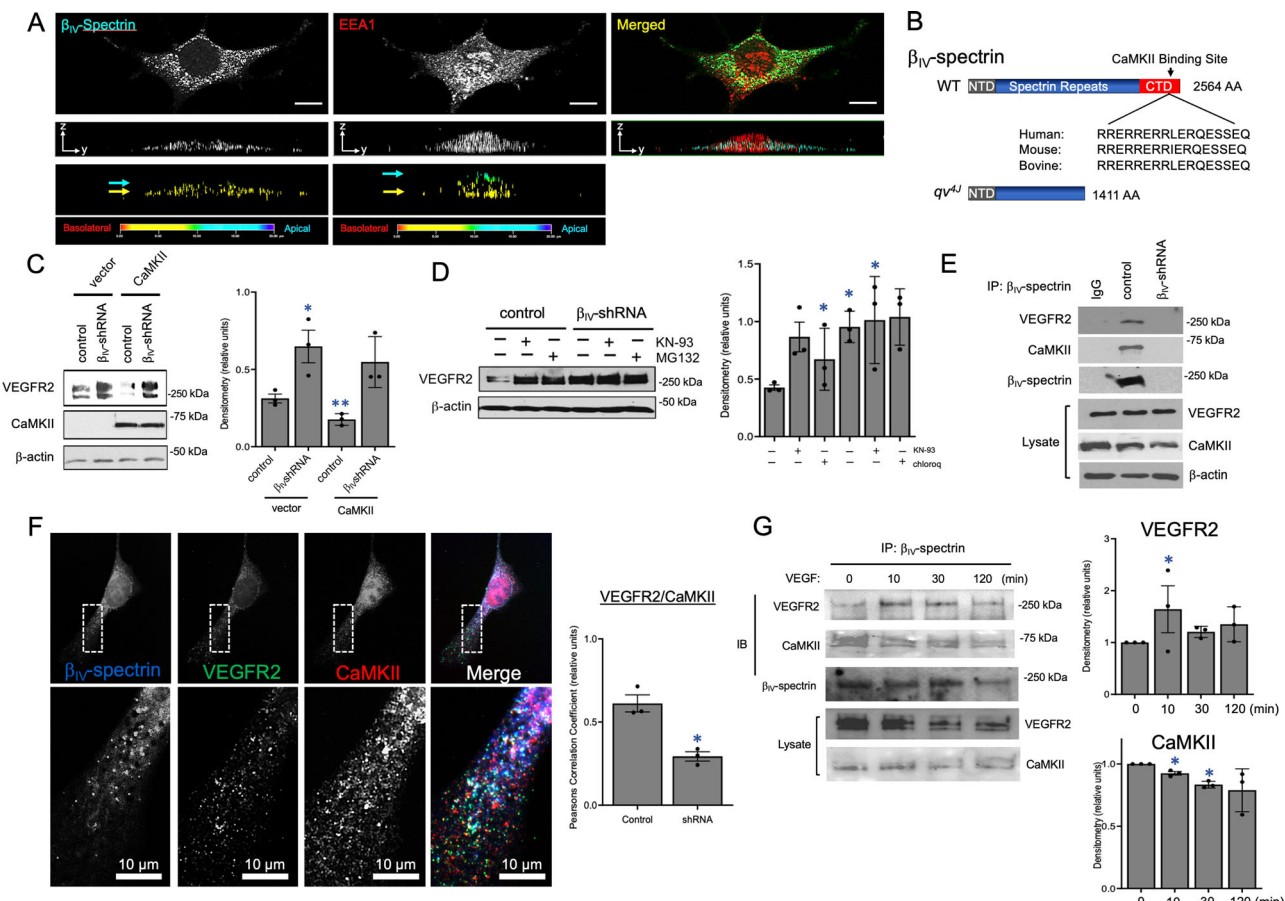

**Fig. 4 β<sub>IV</sub>-spectrin recruits CaMKII to form a trimeric complex with VEGFR2 to promote VEGFR2 turnover. A** Confocal Z-stack image analysis shows endogenous β$_{IV}$-spectrin localization in punctate clusters (green) that localize at the basolateral membrane (yellow arrows and clustered staining correspond to basolateral polarity in topology heat map). EEA1 (red) shows both membrane and cytoplasmic localization as indicated by green and yellow arrows and topology heat map. **B** Schematic of WT and β$_{IV}$-spectrin mutant, qv4J, which lacks the C-terminal domain containing CaMKII-binding site. The peptide sequence showing the precise CaMKII-binding site is conserved. **C** Western analysis of total VEGFR2 and CaMKII expression in control and β$_{IV}$-shRNA MEECs upon transfection with vector control or CaMKIIα. Densitometry quantification representative of three independent experiments, where data is presented as mean values ± SEM. A Type 2 $t$-test results show: $*p = 0.03$, $**p = 0.01$ relative to vector control. Source data are provided as a Source data file. **D** Western analysis shows total VEGFR2 levels in WT and β$_{IV}$-shRNA MEECs upon treatment with KN-93 (8 μM) or chloroquine (50 μM) for 2 h. Densitometry quantification represented as mean values ± SEM of three independent experiments. A Type 2 $t$-test results show: $*p = 0.03$ or lower, relative to no treatment control. Source data are provided as a Source data file. **E** Endogenous immunoprecipitation of β$_{IV}$-spectrin results in co-IP of endogenous VEGFR2 and CaMKII in control but not β$_{IV}$-shRNA MEECs. Mouse IgG used as negative control. **F** Immunofluorescence tri-staining shows endogenous β$_{IV}$-spectrin (red), VEGFR2 (green), and CaMKII (blue) and merged (white) in MEECs. White arrows indicate sites of co-localization. Quantification based on Pearsons correlation coefficient. Graph represents the average of three independent experiments where at least 12 cells were quantified per group per experiment presented as a mean value ± SEM. A Type 2 $t$-test results show: $*p = 2.27E^{-17}$ over control cells. Source data are provided as a Source data file. **G** Representative biochemical experiment in which endogenous β$_{IV}$-spectrin is immunoprecipitated and probed for endogenous VEGFR2 and CaMKII upon treatment with VEGF (50 ng/mL) for the indicated time points (0–120 min). Graph represents densitometry quantification of $n = 3$ independent experiments. Data are presented as a mean value ± SEM normalized to 0 min time point. A Type 2 $t$-test results show: $*p = 0.04$ or lower relative to 0 min. Source data are provided as a Source data file.

competitive positioning, here our results suggest that a select group of ECs that express β$_{IV}$-spectrin are predisposed to becoming stalk cells by directly limiting VEGFR2 expression and downstream signaling.

While β$_{IV}$-spectrin ultimately functions as a powerful suppressor of tip cells, this function may be largely independent of the hallmark Dll4/Notch signaling axis for three reasons. First, we observed that VEGFR2-downregulation is associated with phosphorylation-induced receptor degradation rather than Notch-mediated VEGFR2 gene repression. Second, the VEGFR2 transcript level proved higher in control rather than in β$_{IV}$-spectrin deficient cells despite their opposite protein levels, which not only precludes Notch-mediated gene repression but also argues against β$_{IV}$-spectrin being a mere fine-tuning mechanism of VEGFR2 turnover. Third, we further

observed that β$_{IV}$-spectrin does not correlate strongly with Notch signaling, as VEGFR1 expression was actually higher in β$_{IV}$-sRNA cells (Fig. 7A). Similarly, we did not find evidence that β$_{IV}$-spectrin controls TGF-β or BMP9 signaling, which are known pathways that promote stalk cell gene expression. Although an indirect coordination with the Dll4/Notch/Jagged1 feedback loop or the Smad pathways cannot be ruled out, our data nevertheless suggest a stalk cell-intrinsic role for β$_{IV}$-spectrin in limiting VEGFR2 levels which yields a more permissive environment for others to gain initial tip-cell advantage. Naturally, this concept prompts prospective studies to identify what governs β$_{IV}$-spectrin expression during vascular sprouting, and its pathogenic roles in cancer or pro-inflammatory and oxidative stress-related vascular conditions.

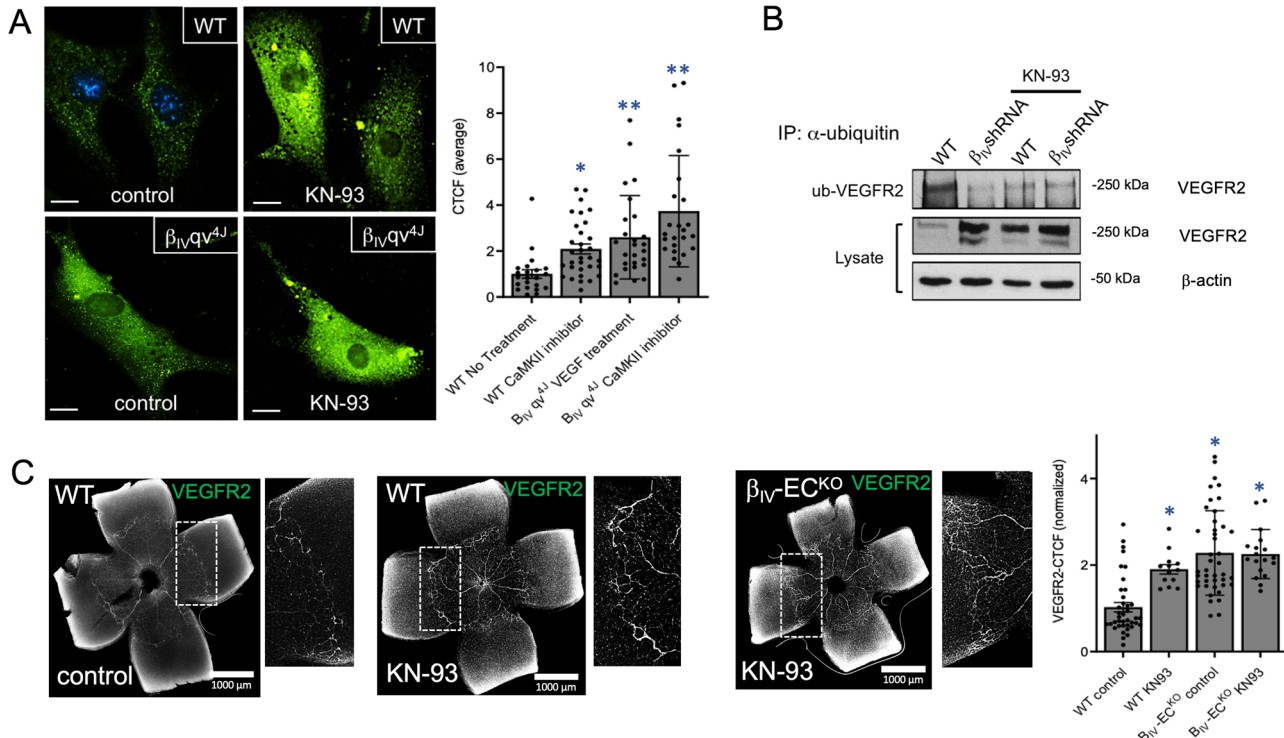

**Fig. 5 $\beta_{IV}$-spectrin mediates CaMKII-induced VEGFR2 internalization and turnover. A** Immunofluorescence images indicate cell surface levels of endogenous VEGFR2 in nonpermeabilized control and $\beta_{IV}qv^{4J}$ ECs upon treatment with KN-93 (2 µM for 4 h). Scale bar: 20 µm. Data is presented as a mean normalized value ± SEM, where $n = 22, 32, 26, 25$ cells based on 3 independent experiments for WT No treatment, WT CaMKII inhibitor, $\beta_{IV}qv^{4J}$ VEGF treatment, and $\beta_{IV}qv^{4J}$ CaMKII inhibitor, respectively. A Type 2 $t$-test results show: $*p = 0.0001$, $**p = 0.0001$ or lower compared to WT. Source data are provided as a Source data file. **B** Western analysis shows immunoprecipitation of total ubiquitin followed by immunoblotting for VEGFR2 in WT versus $\beta_{IV}$-shRNA MEECs upon treatment with KN-93 (2 µM 4 h). Data representative of three independent experiments. **C** VEGFR2 immunofluorescence staining of P5 WT and $\beta_{IV}$-EC$^{KO}$ retina upon simultaneous tamoxifen induction and KN-93 treatment via IP injection at P1 and P3. Graph quantification based on average normalized CTCF values ± SEM, where $n = 39, 13, 42, 18$ regions of interest in 3 separate mouse retinas for WT control, WT KN-93, $\beta_{IV}$-EC$^{KO}$ control, $\beta_{IV}$-EC$^{KO}$ KN-93, respectively. A Type 2 $t$-test results show: $*p = 5.34E^{-7}$ or lower compared to P5 WT. Source data are provided as a Source data file.

But aside from the disease implications, the discovery of $\beta_{IV}$-spectrin expression in the endothelial system is also of molecular significance since this cytoskeletal protein often exists as a heterotetramer with other α- and β-spectrin subunits, each with its distinct set of membrane cytoskeletal binding components and signaling properties as defined in other cell types. And although it appears that the N-terminal domain of $\beta_{IV}$-spectrin is dispensable as the C-terminal truncation ($\beta_{IV}$-qv$^{4J}$) yielded similar vascular and VEGF signaling defects as the knockout ($\beta_{IV}$-EC$^{KO}$), there may be important structural and signaling roles yet to be discovered for the N-terminal domain, and likewise, additional C-terminal functions that are CaMKII-independent. We also note that, while VEGFR1 expression was not changed by $\beta_{IV}$-spectrin, our quantitative proteomics analysis revealed hundreds of other candidate proteins potentially impacted by this cytoskeletal protein, including the insulin receptor which was highly upregulated (Figure S3). How this receptor and many others tie into $\beta_{IV}$-spectrin regulation of sprouting angiogenesis or vascular maintenance, and whether they directly or indirectly associate with this cytoskeletal complex, remain to be determined.

The fact that CaMKII-induced VEGFR2-S984 phosphorylation, enhanced receptor internalization, and degradation all strictly require functional $\beta_{IV}$-spectrin expression strongly support our conclusion that this membrane cytoskeletal protein is essential for the local recruitment and targeting of the kinase to the plasma membrane for substrate access. Accordingly,

there was a notable reduction in ERK activation associated with $\beta_{IV}$-spectrin dysfunction in all three EC types ($\beta_{IV}$-shRNA, $\beta_{IV}$-qv$^{4J}$, and $\beta_{IV}$-EC$^{KO}$ ECs), indicating that enhanced VEGFR2 membrane retention impairs ERK signaling. However, it is unclear why other downstream pathways such as PLCγ, which reportedly requires VEGFR2 endocytosis for efficient signal propagation, were not impaired (Fig. 3B–D; Figure S7C). Given that our work mostly focused on VEGFR2 regulation at basal state (i.e., standard EC media containing growth factors including VEGF), the loss of $\beta_{IV}$-spectrin likely impairs but does not prevent endocytic recycling of the receptor. While our biochemical and immunofluorescence studies provided qualitative assessments, future studies should focus on individual pathways to determine which of the VEGFR2 downstream pathways are most strongly tied to $\beta_{IV}$-spectrin/CaMKII function.

There are numerous receptors, chaperones, and ubiquitin ligases involved in VEGFR2 trafficking including c-Cbl, Ephrin-B2, synectin, and integrins, among others[32,37–40]. Based on our results, follow-up studies should explore detailed mechanisms of how CaMKII-induced VEGFR2-S984 phosphorylation allows for coordination with these molecular chaperones. Nonetheless, identification of this serine-specific site is significant as it marks one of the few instances of a serine/threonine kinase directly targeting VEGFR2 function. As mentioned, the only other major instance involves PKC-induced VEGFR2 phosphorylation and degradation although the exact phosphosite has not been fully

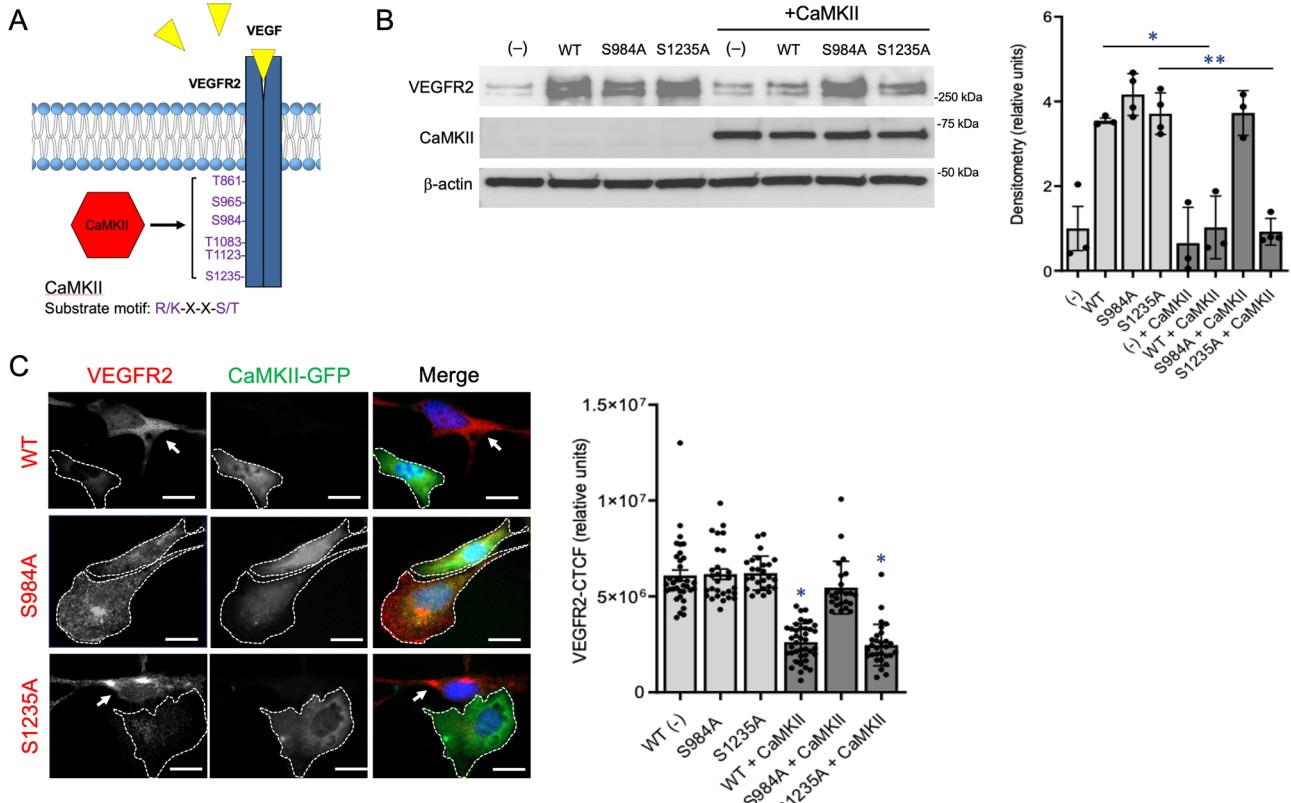

**Fig. 6 β_IV-spectrin recruits CaMKII to the membrane to promote phosphorylation-induced VEGFR2 turnover. A** Schematic shows β_IV-spectrin-mediated CaMKII potentially targeting one or more of its consensus R/K-X-X-S/T phosphorylation motifs present on VEGFR2. **B** Representative western shows total VEGFR2 and CaMKII levels in MEECs transiently overexpressing either WT, VEGFR2 point mutant S984A, or S1235A with CaMKII. Graph densitometry quantification indicates mean values ± SEM of where $n = 3$ independent experiments. A Type 2 $t$-test results show: $*p = 0.005$ or lower and $**p = 0.01$ or lower when compared to the control or as indicated. Source data are provided as a Source data file. **C** Immunofluorescence images show cell surface levels of ectopic expression of VEGFR2 WT, S984A or S1235A (red) in the presence or absence of CaMKII (green) in nonpermeabilized MEECs. Scale bar: 50 μm. Graph indicates average normalized CTCF values ± SEM, where $n = 35, 28, 26, 40, 21, 29$ cells per experiment for WT (−), S984A, S1235A, WT + CaMKII, S984 + CaMKII and S1235A + CaMKII, respectively, derived from 3 independent experiments. A Type 2 $t$-test results show: $*p = 1.62E^{-15}$ or lower when compared to WT (−). Source data are provided as a Source data file.

mapped[18]. It will be important to investigate whether the β_IV-spectrin/CaMKII complex and PKC are interdependent given their crosstalk in other systems[41].

While beyond the scope of the present study, a fundamentally elusive aspect of our work relates to why an endothelial-specific β_IV-spectrin knockout results in partial embryonic lethality whereas global knockout mice are viable notwithstanding various neuropathies and myopathies that increase in severity with age. An EC-specific deletion may be more lethal because of compensatory changes during embryogenesis, whether through direct or indirect effects on other spectrin family members, that occur in the global knockout model. Clearly, further studies are warranted to address these questions. But with specific regards to our inducible β_IV-EC^{KO} mice, we employed various intercrossing strategies to improve embryonic viability, but the overall yield of neonates proved consistently lower than Mendelian expectations relative to the Cre-positive WT mice. Toxicities associated with tamoxifen were not a major determinant of embryonic lethality as the dosage was extensively optimized. We therefore conclude that unknown factors as second hits are necessary to cause irreparable defects in β_IV-spectrin deficient embryos. In addition to exploring these aspects, our ongoing work also aims to address how β_IV-spectrin deficiency throughout development influences vascular aging as well as its pathogenic roles during high-fat diet-induced retinopathy.

In summary, here we define important facets of spectrin biology as results show that β_IV-spectrin, a highly specialized cytoskeletal protein previously studied in neuronal and cardiac contexts, is expressed in a spatially dynamic manner in sprouting vessels. We find that β_IV-spectrin preferentially targets VEGFR2 expression in stalk cells through a distinct phosphorylation-dependent mechanism of receptor turnover. Collectively, these results support β_IV-spectrin as a critical mediator of stalk cell specification and provide molecular insights to inform unique therapeutic approaches to target VEGF signaling.

## Methods

**β_IV-spectrin knockout and mutant mice.** qv^{4J} and β_IV-spectrin fl/fl mice are C57/Bl6 and were generous gifts from Dr. Thomas Hund (Ohio State University). Endothelial-specific Cre mouse model (Cdh5(PAC)-CreERT2) was purchased from Taconic. To generate and induce β_IV-spectrin knockout in β_IV-EC^{KO} mice, β_IV-spectrin fl/fl and Cdh5(PAC)-CreERT2 lines were first crossed and genotyped. All experiments were performed using male and female mice at age P5. For embryo studies, tamoxifen was administered via oral gavage (0.12 mg/g body weight) of the pregnant mice on E8.5, then embryos isolated at the indicated time points. All animal procedures were performed in accordance with the guidelines approved by the University of Arizona Institutional Animal Care and Use Committee.

**Isolation of endothelial cell from mice.** Primary endothelial cells were isolated from mouse lung, heart, liver, and kidney. Tissues were finely chopped then transferred to warm collagenase solution (1 mg/ml collagenase in 1X DMEM with

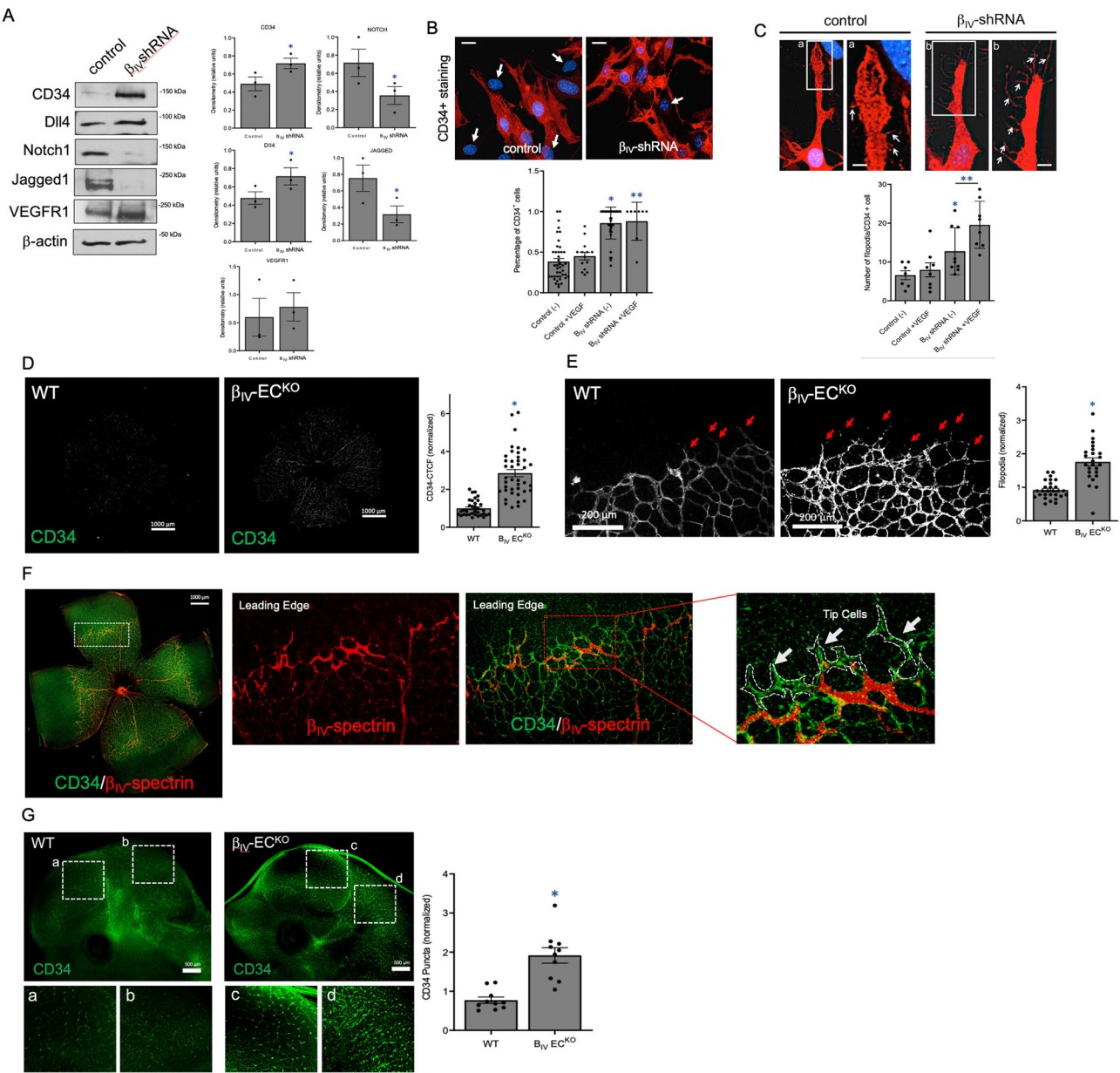

**Fig. 7 $\beta_{IV}$-spectrin acts as a Notch-independent regulator of tip-cell properties in stalk cells. A** Representative westerns show comparative levels of tip and stalk cell markers in control versus $\beta_{IV}$-shRNA MEECs. Graph quantifications represented as mean values ± SEM of 3 independent experiments. A Type 2 $t$-test results show: *$p = 0.04$ relative to control. Source data are provided as a Source data file. **B** Immunofluorescence analysis indicates CD34 + staining counts in control versus $\beta_{IV}$-shRNA ECs in the presence or absence of VEGF stimulation for 4 h. Scale bar: 20 µm. Graph shows the percentage of total CD34 + counts per 100 cells in each group presented as mean value ± SEM. A Type 2 $t$-test results show: *$p = 5.77E^{-15}$, **$p = 7.96E^{-8}$ relative to control no treatment. Source data are provided as a Source data file. **C** Immunofluorescence analysis demonstrates the formation of filopodial projections in control versus $\beta_{IV}$-shRNA ECs upon VEGF stimulation for 2 h. Scale bar: 5 µm. Graph indicates average number of filopodial projections per CD34 + cells ± SEM, where $n = 7, 8, 9, 8$ regions of interest in 3 separate mouse retinas for the Control (−), Control+ VEGF, $\beta_{IV}$-shRNA (−), and $\beta_{IV}$-shRNA+ VEGF, respectively. A Type 2 $t$-test results show: *$p = 0.03$ relative to control no treatment, **$p = 0.01$ as indicated. Source data are provided as a Source data file. **D** Immunofluorescence images show CD34 staining in P5 retina of WT and $\beta_{IV}$-EC$^{KO}$ mice. Graph quantification based on normalized CTCF values ± SEM, where $n = 34$ and 41 for WT and $\beta_{IV}$-EC$^{KO}$, respectively, from 6 separate mice retinas per group. A Type 2 $t$-test results show: *$p = 0.01$ relative to WT. Source data are provided as a Source data file. **E** Higher magnification imaging of CD34 shows filopodial projections at the leading edge of the vascular plexus of WT and $\beta_{IV}$-EC$^{KO}$ retina at P5. Graph represents percentage of filopodial counts normalized to WT ± SEM, where $n = 28$ and 25 for WT and $\beta_{IV}$-EC$^{KO}$, respectively from 5 separate mouse retinas. A Type 2 $t$-test results show: *$p = 1.87E^{-5}$. Source data are provided as a Source data file. **F** Immunofluorescence co-staining reveals that $\beta_{IV}$-spectrin expression is excluded from the leading-edge tip cells (white dotted line and arrows in magnified image). **G** Representative immunofluorescence images and graph indicate greater CD34 punctate staining in $\beta_{IV}$-EC$^{KO}$ than WT embryos at E12.5. Each value was normalized by the mean value of the WT value and data are presented as mean values ± SEM, where $n = 10$ regions of interest from 4 separate embryos. A Type 2 $t$-test results show: *$p = 0.01$ relative to WT. Source data are provided as a Source data file.

Filter with 0.2 µm syringe filter) for enzymatic digestion for 45 min at 37 °C. Digested tissues were passed through a 15 g needle 10 times before being strained through a 70 mm cell strainer. Strained and washed tissue solutions were centrifuged at 4 °C @ 1000 rpm for 5 min. The pellet was resuspended in 0.1% BSA in PBS and incubated with CD31 antibody-conjugated Dynabeads for 15 min at room temperature with gentle shaking then washed with 0.1% BSA/PBS 5×. The beads were resuspended in 1 mL of MEECs Media (MCDB-131 Medium (Gibco) with 10% (v/v) Fetal Bovine Serum, 2 mM L-glutamine (Gibco), 1 mM sodium pyruvate (Gibco), 100 µg/ml of heparin (Sigma-Aldrich) and endothelial cell growth supplement (Sigma-Aldrich)) with 1x penicillin streptomycin and plated on gelatin-coated cell-culture flasks. Upon 2–3 days, cells were separated from beads with trypsin (Gibco).

**Isolation of retina, fluorescence staining, and analysis.** Eyes were fixed for 15 min in 4% paraformaldehyde on ice then rinsed 2× with PBS. Upon fixing, the optic nerve and surround tissues were removed, and retina dissected with 4 radial incisions, then incubated in methanol at −20 °C overnight. Methanol was rinsed out in PBS then blocked (5% goat serum, 0.2% BSA, 0.3% Triton X-100 in 1X PBS) for 1 h at RT. Retina was incubated with primary Ab (1:250 ratio) in 100 µl blocking solution at 4 °C OVN, then washed for 15 min 3× with 0.3% Triton X-100 in PBS. Retina was incubated in secondary at 4 °C OVN, then washed 4× for 30 min with 0.3% Triton X-100 in PBS. Retina was mounted in anti-fade onto the coverslips and images acquired using Nikon A1R confocal microscope prior to analysis with ImageJ/FIJI program.

**Embryos.** Mice embryos were fixed with 2% paraformaldehyde in PBS for 30 min on ice, then washed in PBS prior to incubation in 50% methanol/PBS for 10 min, 75% methanol/PBS for 10 min, then 100% methanol for storage at −20 °C OVN. Embryos were then rehydrated in 50% methanol for 10 min, and then washed 3× in PBS. Embryos were blocked using blocking buffer (2% skim milk, 0.2% BSA, 0.3% Triton X-100 in PBS) for 2 h at room temperature and embryos were incubated with primary antibody (CD31) in blocking buffer at 4 °C overnight. Following day, embryos were washed 3× with blocking buffer at 4 °C for 1 h then washed 2× for 1 h at RT. Following primary antibody incubation, embryos were incubated with secondary fluorophore antibody at 4 °C OVN, then washed 3× for 1 h with blocking buffer, then final washing with 0.3% Triton X-100 in PBS for 30 min at RT. The embryos were mounted using anti-fade onto the glass slides with coverslips. Relative CD31 levels and images were analyzed using ImageJ/FIJI and Adobe Photoshop.

**Cell culture and transfection.** Mouse embryonic endothelial cells were cultured in MCDB-131 (GIBCO) supplemented with 10% (v/v) fetal bovine serum, 2 mM L-glutamine (Gibco), 1 mM sodium pyruvate (Gibco), 100 µg/ml of heparin (Sigma-Aldrich), and endothelial cell growth supplement (Sigma-Aldrich). Cells were maintained in T-75 culture flasks in a 37 °C incubator with 5% $CO_2$ and were passaged every 2–3 days upon reaching 80–90% confluence. Transfection was performed using 1.5 ratio of Lipofectamine 2000 Transfection Reagent (Thermo Fisher) in Opti-MEM Reduced Serum Medium (Gibco) for 6 h and replace regular cell-culture medium for over 30 h.

**Spectrin $\beta_{IV}$-shRNA knockdown stables cell line.** Mouse embryonic endothelial cells (MEECs) were transfected using scrambled control shRNA plasmid or $\beta_{IV}$-spectrin targeting human and mouse shRNA plasmids (Sigma-Aldrich Mission shRNA) or mouse shRNA lentivirus (Santa Cruz Biotec) with 5 µg/ml Polybrene in regular medium. sh-$\beta_{IV}$ Spectrin knockdown stables cells were selected with puromycin (2 mg/mL) in regular cell-culture medium. then colonies were isolated and biochemically validated for $\beta_{IV}$-spectrin knockdown.

**Immunoprecipitation.** Cells were washed three times with 1X PBS, lysed on ice with lysis buffer (20 mM HEPES pH 7.4, 150 mM NaCl, 2 mM EDTA, 10 mM NaF, 10% (v/v) Glycerol, and 1% (v/v) NP-40) with phosphatase and protease inhibitors (1:1000) for 15 min and stabilization for 15 min on ice. Then, Centrifugation at 13,000 rpm for 10 min. Supernatants were incubated with appropriate antibodies for 6 h and protein agarose G or A beads for overnight at 4 °C. Immunoprecipitants were then pelleted and washed three times using the lysis buffer before storing them with 2 × SDS sample buffer followed by western blot analyses.

**Western blotting.** Cell lysates were separated by SDS-PAGE and electrophoretic transferred onto the PVDF (Polyvinylidene difluoride) membranes (BIO-RAD). Transferred membranes were blocked with 5% skim milk in TBS with 0.1% Tween-20, and then incubated with primary antibodies at 4 °C overnight. Following day, membranes were washed three times in TBS buffer with 0.1% Tween-20 and incubated with the secondary antibody for 45 min at room temperature. Membranes were washed five times in TBS buffer with 0.1% Tween-20 each 5 min then imaging by ChemiDoc Imaging system (BIO-RAD). All antibodies used are described in Supplementary Data 1.

**Immunofluorescence.** Cells grown onto the gelatin (Sigma-Aldrich) coated coverslips in 6-well plates were fixed with 4% paraformaldehyde for 15 min, blocked with 5% bovine serum albumin (BSA) in 1X PBS with 0.3% Triton X-100 for 1 h. The primary antibody for overnight at 4 °C with 1% BSA and 0.3% Triton X-100 in 1X PBS and then, washed three times with 1X PBS for 5 min and fluorescently conjugated secondary antibodies were incubated at room temperature for 1 h. The coverslips were mounted using anti-fade that contained 4',6-diamidino-2-phenylindole. Images were obtained via Nikon A1R confocal microscope or Zeiss Axio7/Apotome2 inverted fluorescence microscope with a monochrome camera.

**Endothelial capillary-sprouting assays.** Matrigel (BD Biosciences) was plated in 12-well plates and allowed to polymerize at 37 °C for 30 min. MEECs were plated ($1.5 \times 10^5$ cells) in MCDB-131 media and grown for 16 h. For each well, digital images (×10 magnification) of random fields were acquired, and capillary branches were counted in each field. Experimental condition was normalized as percentage compared to WT.

**Crystal violet cell growth assay.** ECs were plated at 15,000 in 12-well plates OVN. Cells were fixed at indicated time points (4% paraformaldehyde in PBS for 15 min). Following fixation, cells were washed, then stained with 0.1% crystal violet for 20 min. Cells were washed 3×, then air-dried for 30 min prior to destaining (10% acetic acid, 50% methanol, 40% water) for 20 min, and absorbance readings taken at 590 nm in a microplate reader.

**Transwell migration assay.** ECs were seeded in the upper chamber of a transwell filter in complete growth media, coated both at the top and bottom with gelatin. Cells were allowed to migrate for 16 h toward the lower chamber. Cells that migrated to the bottom surface of the filter were fixed, stained, then digitally imaged and counted.

**In-gel digestion.** To determine changes in EC proteome upon $\beta_{IV}$-spectrin depletion, cell lysates were resolved on a 10% SDS-PAGE gel and stained with Bio-Safe Coomassie G-250 Stain. Each lane of the SDS-PAGE gel was cut into five slices. The gel slices were subjected to trypsin digestion and the resulting peptides were purified by C[18]-based desalting exactly as previously described[42,43].

**Mass spectrometry and database search.** HPLC-ESI-MS/MS was performed in positive ion mode on a Thermo Scientific Orbitrap Fusion Lumos tribrid mass spectrometer fitted with an EASY-Spray Source (Thermo Scientific, San Jose, CA). NanoLC was performed exactly as previously described[42,43]. Tandem mass spectra were extracted from Xcalibur 'RAW' files and charge states were assigned using the ProteoWizard 3.0 msConvert script using the default parameters. The fragment mass spectra were searched against the 2020 Mus musculus Swissprot database (17,063 entries) using Mascot (Matrix Science, London, UK; version 2.4). For the detection of $\beta_{IV}$-spectrin expression in proliferating ECs (Figure S1B), the fragment mass spectra were searched against the 2020 Mus musculus TrEMBL database (69,490 entries) using Mascot (Matrix Science, London, UK; version 2.4). The search variables that were used were 10 ppm mass tolerance for precursor ion masses and 0.5 Da for production masses; digestion with trypsin; a maximum of two missed tryptic cleavages; variable modifications of oxidation of methionine and phosphorylation of serine, threonine, and tyrosine. Cross-correlation of Mascot search results with X! Tandem was accomplished with Scaffold (version Scaffold_4.8.7; Proteome Software, Portland, OR, USA). Probability assessment of peptide assignments and protein identifications were made using Scaffold. Only peptides with ≥95% probability were considered. Label-free peptide/protein quantification and identification. Progenesis QI for proteomics software (version 2.4, Nonlinear Dynamics Ltd., Newcastle upon Tyne, UK) was used to perform ion-intensity based label-free quantification as previously described[44]. To determine global differences in protein phosphorylation abundance between WT and $\beta_{IV}$-spectrin depleted cells, 100 mg of protein lysate per sample ($n = 3$) was subjected to in-solution tryptic digestion and phosphopeptide enrichment using sequential enrichment from metal oxide affinity chromatography per manufacturer's protocol (Thermo Scientific, San Jose, CA).

**Statistics and reproducibility.** Statistical analysis was performed using an unpaired two-tailed student $t$-test. Significant statistical differences between groups were indicated as: *$P < 0.05$. Data are presented as mean ± SEM. No statistical method was used to predetermine sample size. No data was excluded from the analyses. The experiments were not randomized. The investigators were not blinded to allocation during experiments and outcome assessment. Statistical analyses and graphics were carried out with GraphPad Prism software and Microsoft Excel.

**Reporting summary**. Further information on research design is available in the Nature Research Reporting Summary linked to this article.

## Data availability

The proteomics data generated in this study have been deposited in the ProteomeXchange via the PRIDE database under accession code PXD026618. The proteomics data generated in this study are provided in the Supplementary Information/ Source data file. Source data are provided with this paper.

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

## Acknowledgements

We would like to thank the University of Arizona Cancer Center for their assistance. This work was supported in part by NIH grant GM128055 awarded to N.Y.L., the University of Arizona Cancer Center internal funding, Cancer Center Support Grant P30CA023074 and P30DA051355.

## Author contributions

E.K. and C.C.P. conducted the majority of the experiments and analysis as primary authors; E.K. contributed to the writing of the paper. A.R. performed retinal staining and data analysis. S.K. performed the RT-qPCR experiments. P.C.F., T.A., and H.R.O. performed the biochemical experiments. J.J.L. and G.M. assisted with confocal imaging and analysis. P.R.L. assisted with all MS proteomics work. T.G.G., T.J.H., and P.J.M. provided the β_IV-spectrin flox and mutant mice. N.A.E., Y.S.L., T.W.V., T.L.M., and K.M. assisted with experimental procedures and edited the paper. N.Y.L. conceived the study and edited the paper with contributions from E.K.

## Competing interests

The authors declare no competing interests.
