## [Peer Review File · Nature Communications]

β IV-spectrin as a stalk cell-intrinsic regulator of VEGF signalingReviewers' Comments:

Reviewer #1:

Remarks to the Author:

In this paper, Kwak et al. identified betaIV-spectrin as a new regulator of sprouting angiogenesis. Genetic deletion of the gene in mice resulted in the mis-patterning of blood vessels as well as an increase in vessel density. This vascular phenotype arose due to increased VEGFR2 expression and activity after betaIV-spectrin depletion in ECs. The authors further demonstrated that VEGFR2 turnover depends on betaIV-spectrin regulation of CaMKII localisation to the plasma membrane, where it phosphorylates VEGFR2 at a newly identified phosphorylation site (Ser984) that regulates VEGFR2 internalisation and degradation.

While it is evident that the deletion of betaIV-spectrin in ECs in mouse results in vascular defects (increased EC number and vessels density), the data presented does not convince me that betaIV-spectrin regulates tip/stalk cell specification. This stems from the fact that there are serious issues about the in vivo data (highlighted in the point-by-point concerns below) that need to be addressed.

Major concerns:

1) There are flaws in the interpretation of the zebrafish data.

- i. There is a mistake in the staging of the embryos. In Fig. 1E, the authors write that they analysed the embryos at 12hpf. However, ISVs only begin to form at around 20-22hpf. The embryos in the images in the top panels of Fig. 1G look older than 24 hpf. This needs to be corrected.
 - ii. The quality and the resolution of the images are too poor for making conclusions. For example, the authors state that there are hypersprouting defects in ISVs at "24 hpf". To me, the arrows in the image in Fig. 1G show ISVs in the contralateral side of the embryo. Also, the authors did not quantify the hypersprouting that they claim to happen.
 - iii. Was the specificity of the morpholino verified? Can the authors rescue the phenotype? Was splicing efficiently blocked? The authors did not provide data on these experiments.
- In conclusion, it is difficult to believe the zebrafish data presented.

2) I have doubts about the specificity of the immunostaining for betaIV-spectrin and VEGFR2 proteins in the retina and therefore the conclusions that the authors make on the increase in VEGFR2 expression in stalk cells after betaIV-spectrin deletion.

- i. In cultured ECs, betaIV-spectrin is localised in vesicles. If this is also true in vivo, then it will be very difficult to detect betaIV-spectrin in ECs of the retina at the magnification and resolution that the images are shown. In fact, the immunostaining looks very luminal, especially in the images in Fig. 7F. Is the anti-betaIV-spectrin antibody labelling erythrocytes? That may explain why the authors "observe betaIV-spectrin in stalk cells" since tip cells are not yet lumenised. Also, there is strong staining in the big vessels of the retina but I cannot determine whether it is the artery or the vein since there isn't an endothelial counterstain in Fig. 1E.
- ii. The immunostaining for VEGFR2 in the retina looks very different to that shown in Nakayama et al., 2013, NCB and Benedito et al., 2012, Nature. In these papers, VEGFR2 staining is weak with no clear enrichment at the sprouting front of the retina vessels such that its detection requires confocal imaging. I therefore question whether the signals that the authors detect are specific. The interpretation of the data will be greatly improved by performing confocal imaging.

3) In this paper, the authors relied on the use of CD34 as a marker of EC tip cells in vivo. However, judging from the staining pattern, CD34 expression is not restricted to tip cells, although it is higher in ECs of the migrating front of the retinal vessels. Furthermore, in the 2D cultures (Fig. 7B) where the cells are non-confluent, not organised into a vascular sprout and there are no tip/stalk cells, CD34 is expressed in all ECs. Therefore, the authors should examine other tip cell markers such as Esm1 and Apelin to strengthen their claim that betaIV-spectrin has a role in regulating tip/stalk cell specification, not just regulating the expression level of CD34.

Other comments:

- 1) Figure 1: the number of samples analysed is missing for most panels.
- 2) The authors should mention which endothelial cells were used for MS profiling and the differences between the two MS spectra in Fig. S1A for readers unfamiliar with such data.
- 3) They authors conclude that betaIV-spectrin deficiency promotes EC hyperproliferation (line 155). However, the data they presented to support this conclusion is the quantification cell number (Fig. 2D), which shows an increase EC number in P5 betaIV-ECKO retina. The authors should instead perform BrdU or Ki67 staining to examine proliferation.

Reviewer #2:

Remarks to the Author:

This is an interesting study showing that betaIV spectrin regulates endothelial cell vasculature formation by controlling turnover of VEGFR2, via betaIV spectrin interaction with CAMKII which phosphorylates VEGFR2 at S984. Overall, the work is generally of high quality and convincing, especially the studies on the mouse retinal vasculature. However, the work needs some additional information and some of the experiments are not convincing.

1. Figure 1. The authors state that they used proteomics to identify betaIV spectrin as a candidate upregulated protein in migrating and proliferating endothelial cells, but this is not documented. At the least, mRNA and protein expression data for betaIV spectrin in the types of endothelial cells should be added to Figure 1.
2. Figure 2. Are the variations in embryonic viability in the tam-induced betaIV spectrin KO mouse embryos due to variable Cre expression and/or gene excision?
3. Figure 3. The western blots in the lysosomal inhibition experiment in Figure 3F should be quantified, since this experiment is an important point regarding potential mechanisms for betaIV spectrin regulation of VEGFR2 signaling, and is used by the authors to claim that betaIV spectrin promotes receptor turnover. It would have been interesting to look at VEGFR2 localization under these conditions as well (also see point 4).
4. The locations of betaIV spectrin in the endothelial cells and vasculature are insufficiently explored in the various systems studied here.
 - a. In Figure 1F, why does betaIV spectrin appear to be present in some but not all Fli:EGFP labeled endothelial cells in the zebrafish vasculature? Can the betaIV spectrin staining be improved? Or is it preferentially located in some types of endothelial cells and not others?
 - b. In the retinal vasculature, the authors show more betaIV spectrin in the sprouting vasculature (Figure 2E) and also show more VEGFR2 in the sprouting vasculature of the betaIV spectrin KO retina (Figure 3H-J). They imply that betaIV spectrin is spatially colocalized with VEGFR2 in expanding vasculature in WT retina but they do not directly colocalize VEGFR2 and betaIV spectrin in the endothelial cells. How do betaIV spectrin and VEGFR2 localization compare in quiescent vs sprouting vasculature? This would add to the study.
 - c. It is not at all clear where the betaIV spectrin is located in endothelial cells, nor how this is related to the localization VEGFR2. Figure 4C shows a cell in the left panel immunostained for endogenous betaIV spectrin (the staining is very weak), but it is not clear whether this staining is on the plasma membrane or on internal membranes, such as ER or other types of vesicles. Does VEGFR2 colocalize with endogenous betaIV spectrin on the plasma membrane or in internalized vesicles? How does

betaIV spectrin knockdown affect VEGFR2 localization?

5. Figure 4. The experiments showing betaIV spectrin regulation of VEGFR2 turnover via CAMKII are incomplete and/or misinterpreted. (None of the images have mag bars.)

a. Figure 4A-B comparing expression of the N or C terminal halves of betaIV spectrin on VEGFR2 levels is not convincing. The westerns would need to be repeated and quantified. Figure 4C also does not provide any information about the effects of betaIV spectrin domains on VEGFR2, the purported target of the betaIV spectrin, since VEGFR2 is not localized in the cells. In the cell with overexpressed full length betaIV-spectrin the localization looks very different than the endogenous betaIV-spectrin, like giant bright aggregates or vesicles. What does this mean? The localization of the N terminal or C terminal betaIV spectrins look different again and also do not resemble the endogenous protein (they would need to be colocalized to establish whether they are similar or different from the endogenous full length). These different patterns could all be an artifact of overexpression. This experiment does not provide useful information as to whether betaIV spectrin may regulate VEGFR2 turnover via CAMKII binding, and could be omitted.

b. Figure 4D and E showing a role for CAMKII in VEGFR2 levels are not convincing. The westerns need to be repeated and quantified.

c. Figure 4G colocalizing VEGFR2, betaIV spectrin and CAMKII in an endothelial cell shows large vesicles or aggregates of CAMKII which appear to colocalize with the betaIV spectrin and VEGFR2 in a few locations. However, this is not convincing. Randomly distributed dots may appear to colocalize by chance. The CAMKII appears to be present in large aggregates and this could also lead to artifactual apparent colocalization. The colocalization of two molecules at a time needs to be examined more carefully at a considerably higher magnification to understand what is going on. Furthermore, if the authors want to claim formation of trimeric complexes in cells the immunolocalization needs to be performed at a significantly higher resolution and a FRET colocalization approach is necessary to demonstrate direct complex formation in the cell. Alternatively, the experiment in panel G could be omitted.

6. Figure 7C shows that knockdown of betaIV spectrin in cultured endothelial cells leads to more filopodia formation. However, in the mouse retina, Figure 7F shows that the betaIV spectrin is not in the same cells as the CD34 stained cells at the tip in the expanding vasculature, and is instead in the stalk cells. Where do the filopodia form in the retinal vasculature? How does betaIV spectrin regulate filopodia in retinal endothelial tip cells if it is not present in them? Does the presence of betaIVspectrin in the stalk but not the tip cells lead to different levels of VEGFR2 expression in the two cell types? Some more data and information might clarify the proposed mechanisms of cross-talk between the two endothelial cell types.

Minor comments.

1. Abbreviations need to be spelled out. Please define ERG in the text on line 182 so that the non-specialist can follow the paper. Please define CTCF in the legend to Figure 5A.

2. Many of the images are missing mag bars throughout. Please add them.

3. The sentence structure is often awkward and there are numerous grammatical errors throughout. Sometimes I had to read the sentences twice to understand the points. It is suggested the authors get an English language editor to improve readability.

Reviewer #3:

Remarks to the Author:

In the Nature Communications submission by Kwak and colleagues entitled " β IV-spectrin as a stalk cell-intrinsic regulator of VEGF signaling" the authors describe a previously unknown angiogenic signaling pathway in endothelial cells. Using mass spectrometry profiling of proliferating versus quiescent endothelial cells, the authors identified β IV-spectrin; though, this seems to be quite a simplistic means to find such targets given the complexity of angiogenic responses in the body. In any case, while it is well-established that β IV-spectrin/CaMKII signaling plays an important role in regulating ion channel function in excitable tissues such as the brain and heart as well as in pancreatic β cells, this report outlines a novel role for these proteins in vascular development. In particular, β IV-spectrin was shown to recruit CaMKII to the plasma membrane to phosphorylate VEGFR2 at Ser984, a previously undefined phosphoregulatory site that encourages VEGFR2 internalization and degradation in developing embryonic endothelial cells and in cell cultures. The authors used both Zebrafish and mouse models to examine angiogenic sprouting in development, but not in pathological conditions relevant to human diseases.

They made a number of important observations regarding the identification of β IV-spectrin as a regulator of angiogenic responses, which is novel and interesting. This includes the recruitment of CaMKII to phosphorylate VEGFR2 to control internalization and degradation of this receptor. These are also intriguing findings; however, even though the results generally support the conclusions of the study, enthusiasm for the submission was dampened by the reliance on a limited number experiments and a singular interpretation of the data, which was in some cases not verified through the use of redundant strategies or extensive rigorous controls.

The authors may want to consider the following suggestions and questions to improve their submission:

1. If a number of potentially unique angiogenic markers were identified during the MS screen, what was the criteria for selecting β IV-spectrin for subsequent evaluation? In addition, Figure S1A is difficult to interpret and should be modified.
2. The assembled figures are generally substandard and need to be improved with regard to legibility, organization, and consistency.
3. Ideally, the Zebrafish morpholino (MO) studies shown in Figure 1 should have been validated by comparison to a mutant. While injecting the MO into a mutant provides the most definitive evidence for MO specificity, there are circumstances where a mutant can't be generated. If this is the case, multiple MOs should have been tested and rescue experiments performed (Stainier et al., 2017) to ensure the direct influence of the MO as referred to in the text associated with Figure S1B.
4. Did the authors perform tamoxifen injection in pups to avoid embryonic lethality? It would be interesting to assess angiogenic responses in adult mice with a pathological condition to determine the therapeutic potential of this newly-identified endothelial target.
5. In the Figure 3, comparison of total VEGFR2 and activated receptor levels in WT and β IV-shRNA-treated MEECs, the authors propose a negative feedback response was responsible for the decrease in receptor expression. Characterization of this response should include additional supporting experimentation to show a time course of MG132 treatment. At the same time, the fluorescence imaging of the retinal vasculature could be improved.
6. A more thorough examination of VEGFR2 location in WT and β IV-shRNA cells (i.e., cell surface versus endocytic compartments, recycling endosomes, lysosomes) and turnover would have added to the study.
7. The tissue source of the mouse embryonic endothelial cells (MEECs) used for Figure 3 A to F is a mixture of heart, lung, liver and kidney cells purified using CD31-conjugated beads, which wouldn't account for tissue-specific differences or distinguish between arterial, capillary, and venous cells.
8. What domains are associated with the N-terminal and C-terminal expression constructs? Finer mapping of the potential VEGFR2 regulatory domains would make for a more complete mechanistic study. While the authors point toward a previous study showing CaMKII interaction with the C terminus of β IV-spectrin, a more precise mapping of the interacting domain is warranted.
9. KN-93 has a number of targets other than CaMKII and does not block catalytic activity. As a result, the experiments shown in Figures 4 and S4 would have been more convincing if another means to inhibit activity of this kinase; however, the IP studies somewhat alleviates this concern.

10. The fluorescent co-localization of β IV-spectrin with CaMKII and VEGFR2 is difficult to distinguish at the magnification and resolution presented in Figure 4G.
11. In Figure 6A, the S1235 site is not indicated in the schematic diagram. The information presented in Figure 6C is difficult to interpret based on so few cells. The graph of this data seems to be underpowered from a statistical standpoint. Analysis of 25 cells per group is likely inadequate.
12. The authors rely on presenting what appears to be single experiments for western analyses (e.g., Figure 7A), which should include appropriate statistical analyses of independent experiments and graphical representation.

Reviewer #4:

Remarks to the Author:

This manuscript presents important insights into stalk cell specification by the protein β IV spectrin and provides sound experimental evidence of the mechanism by tracking the phosphorylation dependent association of VEGFR with the β IV spectrin and CaMKII complex and spatial marking of β IV spectrin expression. The overall role of β IV spectrin in tip-stalk cell dynamics is clearly presented and opens doors to many follow-up studies exploring VEGFR2 regulation. My specific comments are as follows:

1. Define/write in full 'VEGF' in line 69 when you first introduce the term.
2. Define/write in full 'PKC' in line 84 when you first introduce the term.
3. Figure S1A- Clearly explain and mark what the top and bottom panels represent. This is the very first figure being discussed, and it's confusing. Is it confluent v/s proliferative or is it neuronal v/s vascular? The caption and text say different things here.
4. Line 127- mention the purpose of adding morpholinos and what it is.
5. Figure 2A- Please write the time points in the figure itself.
6. Line 146,147- If it is still about β IV-ECKO embryos, don't start a new sentence as if it is another experiment. Is it just a different time point or a different embryo?
7. Figure 2F- caption should state that the depletion is induced by tamoxifen.
8. Figure 3A is not readable at all, please increase the size.
9. Figure S3- what are the 6 fractions shown in the Excel sheet (column I)? Are those the gel pieces digested and analyzed separately? If so, why does the method section say that the gel was cut into 5 pieces?
10. In-gel digestion method- The references quoted do not include reduction and alkylation steps prior to trypsin digestion. Is there a reason those were not performed?
11. Line 173- "Instead, β IV spectrin appeared..." this sentence referring to Figure 3F is confusing to the reader, and the caption is also absent in the Figure. Please rephrase the sentence and add a caption.
12. Figure S5- Please provide methods for phosphopeptide analysis. Was any enrichment performed to concentrate phosphopeptides?

Reviewer #1 (Remarks to the Author):

In this paper, Kwak et al. identified betaIV-spectrin as a new regulator of sprouting angiogenesis. Genetic deletion of the gene in mice resulted in the mis-patterning of blood vessels as well as an increase in vessel density. This vascular phenotype arose due to increased VEGFR2 expression and activity after betaIV-spectrin depletion in ECs. The authors further demonstrated that VEGFR2 turnover depends on betaIV-spectrin regulation of CaMKII localisation to the plasma membrane, where it phosphorylates VEGFR2 at a newly identified phosphorylation site (Ser984) that regulates VEGFR2 internalisation and degradation.

While it is evident that the deletion of betaIV-spectrin in ECs in mouse results in vascular defects (increased EC number and vessels density), the data presented does not convince me that betaIV-spectrin regulates tip/stalk cell specification. This stems from the fact that there are serious issues about the *in vivo* data (highlighted in the point-by-point concerns below) that need to be addressed.

Major concerns:

1) There are flaws in the interpretation of the zebrafish data.

i. There is a mistake in the staging of the embryos. In Fig. 1E, the authors write that they analysed the embryos at 12hpf. However, ISVs only begin to form at around 20-22hpf. The embryos in the images in the top panels of Fig. 1G look older than 24 hpf. This need to be corrected.

ii. The quality and the resolution of the images are too poor for making conclusions. For example, the authors state that there are hypersprouting defects in ISVs at “24 hpf”. To me, the arrows in the image in Fig. 1G show ISVs in the contralateral side of the embryo. Also, the authors did not quantify the hypersprouting that they claim to happen.

iii. Was the specificity of the morpholino verified? Can the authors rescue the phenotype? Was splicing efficiently blocked? The authors did not provide data on these experiments.

In conclusion, it is difficult to believe the zebrafish data presented.

We acknowledge that the zebrafish data presented in Figure 1 (originally shown as Fig.1E-G) was weak and thus decided to omit this data as it was an early *in vivo* screening exercise to determine whether a mammalian model was necessary to further pursue this project. We respectfully turn our focus to the real strength of our *in vivo* work in our two mouse models (truncation mutant and EC-specific inducible knockout) along with new imaging data that support a previously unknown role of β_{IV} -spectrin in regulating sprouting angiogenesis via VEGFR2 turnover.

2) I have doubts about the specificity of the immunostaining for betaIV-spectrin and VEGFR2 proteins in the retina and therefore the conclusions that the authors make on the increase in VEGFR2 expression in stalk cells after betaIV-spectrin deletion.

i. In cultured ECs, betaIV-spectrin is localised in vesicles. If this is also true *in vivo*, then it will be very difficult to detect betaIV-spectrin in ECs of the retina at the magnification and resolution that the images are shown. In fact, the immunostaining looks very

luminal, especially in the images in Fig. 7F. Is the anti-betaIV-spectrin antibody labelling erythrocytes? That may explain why the authors “observe betaIV-spectrin in stalk cells” since tip cells are not yet lumenised. Also, there is strong staining in the big vessels of the retina but I cannot determine whether it is the artery or the vein since there isn't an endothelial counterstain in Fig. 1E.

We now provide high-resolution confocal Z-stack images of endogenous β_{IV} -spectrin that show its punctate distribution along the basolateral membrane domains and not in endocytic vesicles when counterstained with EEA1 (endocytic marker).

We respectfully argue that the specificity of the β_{IV} -spectrin retinal staining is accurate since, as originally stated, it is prominently localized to a narrow band of stalk ECs trailing the leading edge of the tip cell populations. If the staining were that of erythrocytes present in lumenized vessels, we would expect to see a more uniform β_{IV} -spectrin staining in all the capillaries throughout the retina. Although, admittedly, we do not understand why there is some basal staining in major arterioles and venuoles, we feel that this does not detract from our principal findings that β_{IV} -spectrin controls angiogenic sprouting and CD34 levels.

ii. The immunostaining for VEGFR2 in the retina looks very different to that shown in Nakayama et al., 2013, NCB and Benedito et al., 2012, Nature. In these papers, VEGFR2 staining is weak with no clear enrichment at the sprouting front of the retina vessels such that its detection requires confocal imaging. I therefore question whether the signals that the authors detect are specific.

The interpretation of the data will be greatly improved by performing confocal imaging. We are confident in our VEGFR2 staining as we have assessed the quality of 4 different commercially available VEGFR2 antibodies before settling on one which we found most suitable, then further optimized the staining with various fixation methods (i.e. methanol vs paraformaldehyde). While higher quality confocal images are also now included, we note that our original VEGFR2 staining images are similarly weak and more diffuse in the control WT. It is only when β_{IV} -spectrin is knocked out that VEGFR2 levels dramatically increase, allowing for greater visualization of its enrichment. This is also entirely consistent with our unbiased MS-proteomics and numerous biochemical assessments.

3) In this paper, the authors relied on the use of CD34 as a marker of EC tip cells in vivo. However, judging from the staining pattern, CD34 expression is not restricted to tip cells, although it is higher in ECs of the migrating front of the retinal vessels. Furthermore, in the 2D cultures (Fig. 7B) where the cells are non-confluent, not organised into a vascular sprout and there are no tip/stalk cells, CD34 is expressed in all ECs. Therefore, the authors should examine other tip cell markers such as Esm1 and Apelin to strengthen their claim that betaIV-spectrin has a role in regulating tip/stalk cell specification, not just regulating the expression level of CD34.

We had to significantly increase the laser intensity in order to capture the filopodial projections of the tip cells when stained with CD34. This is why other parts show some nonspecific staining. However, as suggested by the reviewer, we now show via confocal using Esm1 as another tip cell marker that there are greater number of sprouting tip cells in β_{IV} -spectrin knockout retina (Fig.S9). Additionally, please note that β_{IV} -spectrin is

mostly excluded from the tip cell region near the radial front. With regards to the 2D cultures in Fig.7B, we respectfully note that the original white arrows indicate the DAPI staining of ECs that are not CD34+ and further demonstrated in our graph quantification.

Other comments:

1) Figure 1: the number of samples analysed is missing for most panels.

We now provide this information for all relevant data panels.

2) The authors should mention which endothelial cells were used for MS profiling and the differences between the two MS spectra in Fig. S1A for readers unfamiliar with such data.

A more detailed description is provided in the main text and Fig.S1A modified to improve clarity.

3) They authors conclude that betaIV-spectrin deficiency promotes EC hyperproliferation (line 155). However, the data they presented to support this conclusion is the quantification cell number (Fig. 2D), which shows an increase EC number in P5 betaIV-ECKO retina. The authors should instead perform BrdU or Ki67 staining to examine proliferation.

The combined data that show increased EC cell counts in culture upon β_{IV} -spectrin knockdown (Fig.1E) and EC-specific nuclear staining (ERG) in retina as shown in the original data (Fig.2D) support our conclusion that β_{IV} -spectrin promotes EC hyperproliferation. Indeed, ERG has been previously used as a marker of EC proliferation in vivo (Graeme et al., *Dev Cell*, 2015; Pontes-Quero et al., *Nat Commun*, 2019). While we considered BrdU or Ki67, we also felt that BrdU may also stain for retinal ganglion, neural and photoreceptors.

Reviewer #2 (Remarks to the Author):

This is an interesting study showing that betaIV spectrin regulates endothelial cell vasculature formation by controlling turnover of VEGFR2, via betaIV spectrin interaction with CAMKII which phosphorylates VEGFR2 at S984. Overall, the work is generally of high quality and convincing, especially the studies on the mouse retinal vasculature. However, the work needs some additional information and some of the experiments are not convincing.

1. Figure 1. The authors state that they used proteomics to identify betaIV spectrin as a candidate upregulated protein in migrating and proliferating endothelial cells, but this is not documented. At the least, mRNA and protein expression data for betaIV spectrin in the types of endothelial cells should be added to Figure 1.

We now provide new data showing dynamic β_{IV} -spectrin protein expression at the active log-phase of growth whereas it is dramatically diminished to nearly undetectable levels when cells reach confluence (Fig.1A and graph). This is consistent with in vivo patterns where markedly elevated β_{IV} -spectrin expression is observed mostly in actively proliferating stalk cells near the retinal radial expansion. With regards to mRNA levels,

while our RT-qPCR data is entirely consistent with the westerns, we respectfully request that this be saved for use in our follow-up paper that explores more thoroughly how β_{IV} -spectrin expression is transcriptionally regulated during angiogenesis.

2. Figure 2. Are the variations in embryonic viability in the tam-induced betaIV spectrin KO mouse embryos due to variable Cre expression and/or gene excision?

We have thoroughly genotyped for Cre-positivity in both WT and B4-KO mice and optimized for tam-induced depletion in over 3 independent experiments using freshly isolated ECs from these mice (Fig.2F and graph). Therefore, we believe that the variations are due to unknown second hit insults rather than the efficacy of inducible β_{IV} -spectrin knockout.

3. Figure 3. The western blots in the lysosomal inhibition experiment in Figure 3F should be quantified, since this experiment is an important point regarding potential mechanisms for betaIV spectrin regulation of VEGFR2 signaling, and is used by the authors to claim that betaIV spectrin promotes receptor turnover. It would have been interesting to look at VEGFR2 localization under these conditions as well (also see point 4). We improved on this study by doing a time-course experiment and provide densitometry quantification to demonstrate that β_{IV} -spectrin mediates VEGFR2 turnover.

4. The locations of betaIV spectrin in the endothelial cells and vasculature are insufficiently explored in the various systems studied here.

a. In Figure 1F, why does betaIV spectrin appear to be present in some but not all Fli:EGFP labeled endothelial cells in the zebrafish vasculature? Can the betaIV spectrin staining be improved? Or is it preferentially located in some types of endothelial cells and not others?

As mentioned, we acknowledge that the zebrafish data is weak and therefore decided to omit this data. It was originally used as a quick in vivo screening method, but unfortunately, we no longer have the necessary expertise or zebrafish-related resources to redo this experiment. However, given the depth of our investigation using two mouse models, we believe that our key conclusions are valid.

b. In the retinal vasculature, the authors show more betaIV spectrin in the sprouting vasculature (Figure 2E) and also show more VEGFR2 in the sprouting vasculature of the betaIV spectrin KO retina (Figure 3H-J). They imply that betaIV spectrin is spatially colocalized with VEGFR2 in expanding vasculature in WT retina but they do not directly colocalize VEGFR2 and betaIV spectrin in the endothelial cells. How do betaIV spectrin and VEGFR2 localization compare in quiescent vs sprouting vasculature? This would add to the study.

Despite our assessment of several commercially available VEGFR2 and β_{IV} -spectrin antibodies, the only ones which we found suitable for immunofluorescence staining of retinal vessels happen to be generated from the same species. Therefore, we cannot co-stain for the two proteins in retina (but co-staining is possible for cultured ECs). However, we performed individual staining of mature vessels in adult retina and found moderate levels of VEGFR2 still persisting in β_{IV} -spectrin knockout compared to control (Fig.S4C). As for β_{IV} -spectrin staining, in adult retinal vasculature the staining is very

weak (Fig.S4B). Again, this is consistent with our biochemical analysis of β_{IV} -spectrin expression being readily detectable in actively proliferating but not confluent ECs (Fig.1A) as well as its expression being mostly confined to actively sprouting vessels at the radial front of P5 retina (Fig.2E).

c. It is not at all clear where the betaIV spectrin is located in endothelial cells, nor how this is related to the localization VEGFR2. Figure 4C shows a cell in the left panel immunostained for endogenous betaIV spectrin (the staining is very weak), but it is not clear whether this staining is on the plasma membrane or on internal membranes, such as ER or other types of vesicles. Does VEGFR2 colocalize with endogenous betaIV spectrin on the plasma membrane or in internalized vesicles? How does betaIV spectrin knockdown affect VEGFR2 localization?

As part of new data, we show via confocal Z-stacks that endogenous β_{IV} -spectrin localizes predominantly to the basolateral membrane and not endocytic vesicles such as EEA1 (Fig.4A). VEGFR2 has been previously shown to localize also to the basolateral domains (Hudson et al., *Dev Cell*, 2014). In Fig.4F, we provide new, higher quality images of β_{IV} -spectrin/CaMKII/VEGFR2 colocalization into punctate vesicles (white arrows). Lastly, we provide new confocal Z-stack images of endogenous β_{IV} -spectrin and VEGFR2 (Fig.S5). Notably, we observe that VEGFR2 localizes to basolateral regions in WT control although this basolateral accumulation becomes much greater upon β_{IV} -spectrin knockdown (as indicated by white arrows in β_{IV} -shRNA cells). Combined, these data demonstrate that both β_{IV} -spectrin and VEGFR2 interact mostly at the basolateral compartments and that CaMKII recruitment causes VEGFR2 internalization followed by receptor turnover.

5. Figure 4. The experiments showing betaIV spectrin regulation of VEGFR2 turnover via CAMKII are incomplete and/or misinterpreted. (None of the images have mag bars.)
The scale bars are now included.

a. Figure 4A-B comparing expression of the N or C terminal halves of betaIV spectrin on VEGFR2 levels is not convincing. The westerns would need to be repeated and quantified. Figure 4C also does not provide any information about the effects of betaIV spectrin domains on VEGFR2, the purported target of the betaIV spectrin, since VEGFR2 is not localized in the cells. In the cell with overexpressed full length betaIV-spectrin the localization looks very different than the endogenous betaIV-spectrin, like giant bright aggregates or vesicles. What does this mean? The localization of the N terminal or C terminal betaIV spectrins look different again and also do not resemble the endogenous protein (they would need to be colocalized to establish whether they are similar or different from the endogenous full length). These different patterns could all be an artifact of overexpression. This experiment does not provide useful information as to whether betaIV spectrin may regulate VEGFR2 turnover via CAMKII binding, and could be omitted.

We respectfully agree with the reviewer's assessment that the effects of overexpressing the N and C terminal halves of β_{IV} -spectrin on VEGFR2 levels is not convincing, and ultimately do not provide useful information. While we reperformed this experiment multiple times for quantification during our revision process, upon obtaining high-

resolution confocal Z-stacking, we became convinced that these truncation mutants were misfolding and mislocalizing when compared to our new data that shows endogenous β_{IV} -spectrin localization (new Fig.4A). Hence, as suggested by the reviewer, we decided to omit this study involving overexpression of various β_{IV} -spectrin truncation mutants (originally Fig.4B and C) in favor of more informative 3-D imaging to determine the subcellular localization of endogenous β_{IV} -spectrin, endocytic vesicles (EEA1) and VEGFR2 (Fig.4A,F; Fig.S5). These new data images show that unlike EEA1, β_{IV} -spectrin and VEGFR2 show more basolateral membrane localization. As the reviewer will note, we performed all the subsequent interaction studies at the endogenous level for greater physiologic relevance. We also provide new densitometry quantification (e.g. Fig.4C, D and G).

b. Figure 4D and E showing a role for CAMKII in VEGFR2 levels are not convincing. The westerns need to be repeated and quantified.

Because we omitted the original subfigures Fig.4B-C, Fig.4D and E have now become Fig.4C and D. We have repeated these experiments and now provide densitometry quantifications.

c. Figure 4G colocalizing VEGFR2, betaIV spectrin and CAMKII in an endothelial cell shows large vesicles or aggregates of CAMKII which appear to colocalize with the betaIV spectrin and VEGFR2 in a few locations. However, this is not convincing. Randomly distributed dots may appear to colocalize by chance. The CAMKII appears to be present in large aggregates and this could also lead to artifactual apparent colocalization. The colocalization of two molecules at a time needs to be examined more carefully at a considerably higher magnification to understand what is going on. Furthermore, if the authors want to claim formation of trimeric complexes in cells the immunolocalization needs to be performed at a significantly higher resolution and a FRET colocalization approach is necessary to demonstrate direct complex formation in the cell. Alternatively, the experiment in panel G could be omitted.

While the reviewer kindly allowed us to omit the images in panel F (originally panel G), we have decided to re-do this experiment and show higher quality confocal images showing their subcellular colocalization, as red, green and blue stainings result in white color changes (and unbiased pearsons correlation analysis). But in addition, we show in Fig.4E and 4G that the three proteins interact. Combined, we believe that their moderate co-localization to be credible.

6. Figure 7C shows that knockdown of betaIV spectrin in cultured endothelial cells leads to more filopodia formation. However, in the mouse retina, Figure 7F shows that the betaIV spectrin is not in the same cells as the CD34 stained cells at the tip in the expanding vasculature, and is instead in the stalk cells. Where do the filopodia form in the retinal vasculature? Previous body of work has shown that, in sprouting vasculature, filopodia are produced mostly in tip cells and not in stalk cells. This notion is consistent with our data in which we observe greater number of tip cells and their corresponding levels of VEGFR2-induced filopodia along the leading front when B4 expression is depleted. In essence, this observation supports our conclusion that B4 is selectively expressed in stalk cells to act as a tip cell suppressor.

How does betaIV spectrin regulate filopodia in retinal endothelial tip cells if it is not present in them? Does the presence of betaIVspectrin in the stalk but not the tip cells lead to different levels of VEGFR2 expression in the two cell types? Some more data and information might clarify the proposed mechanisms of cross-talk between the two endothelial cell types. Based on our data we assert that \$\beta_{IV}\$ -spectrin is regulating filopodia by limiting VEGFR2 availability. Indeed, this is consistent with previous findings that VEGFR2 and its downstream signaling induces migration by, among others, enhanced filopodial projections. This is now discussed more in detail in the text.

Minor comments.

1. Abbreviations need to be spelled out. Please define ERG in the text on line 182 so that the non-specialist can follow the paper. Please define CTCF in the legend to Figure 5A.

We apologize for this oversight and now properly define these abbreviations.

2. Many of the images are missing mag bars throughout. Please add them.

The scale bars are now incorporated in the figures.

3. The sentence structure is often awkward and there are numerous grammatical errors throughout. Sometimes I had to read the sentences twice to understand the points. It is suggested the authors get an English language editor to improve readability.

The text has been more thoroughly edited.

Reviewer #3 (Remarks to the Author):

In the Nature Communications submission by Kwak and colleagues entitled " β_{IV} -spectrin as a stalk cell-intrinsic regulator of VEGF signaling" the authors describe a previously unknown angiogenic signaling pathway in endothelial cells. Using mass spectrometry profiling of proliferating versus quiescent endothelial cells, the authors identified β_{IV} -spectrin; though, this seems to be quite a simplistic means to find such targets given the complexity of angiogenic responses in the body.

In any case, while it is well-established that β_{IV} -spectrin/CaMKII signaling plays an important role in regulating ion channel function in excitable tissues such as the brain and heart as well as in pancreatic β cells, this report outlines a novel role for these proteins in vascular development. In particular, β_{IV} -spectrin was shown to recruit CaMKII to the plasma membrane to phosphorylate VEGFR2 at Ser984, a previously undefined phosphoregulatory site that encourages VEGFR2 internalization and degradation in developing embryonic endothelial cells and in cell cultures. The authors used both Zebrafish and mouse models to examine angiogenic sprouting in development, but not in pathological conditions relevant to human diseases.

They made a number of important observations regarding the identification of β_{IV} -spectrin as a regulator of angiogenic responses, which is novel and interesting. This includes the recruitment of CaMKII to phosphorylate VEGFR2 to control internalization and degradation of this receptor. These are also intriguing findings; however, even

though the results generally support the conclusions of the study, enthusiasm for the submission was dampened by the reliance on a limited number of experiments and a singular interpretation of the data, which was in some cases not verified through the use of redundant strategies or extensive rigorous controls.

The authors may want to consider the following suggestions and questions to improve their submission:

1. If a number of potentially unique angiogenic markers were identified during the MS screen, what was the criteria for selecting β_{IV} -spectrin for subsequent evaluation? In addition, Figure S1A is difficult to interpret and should be modified.

The criteria for pursuing β_{IV} -spectrin is now discussed in the main text. Fig.S1A has been modified to make it more intuitive.

2. The assembled figures are generally substandard and need to be improved with regard to legibility, organization, and consistency.

We have now improved the organization, legibility and consistency throughout the manuscript.

3. Ideally, the Zebrafish morpholino (MO) studies shown in Figure 1 should have been validated by comparison to a mutant. While injecting the MO into a mutant provides the most definitive evidence for MO specificity, there are circumstances where a mutant can't be generated. If this is the case, multiple MOs should have been tested and rescue experiments performed (Stainier et al., 2017) to ensure the direct influence of the MO as referred to in the text associated with Figure S1B.

We have decided to omit the zebrafish data as we no longer have the necessary tools or expertise. However, given the robust in vivo data from two mouse models, we stand by our findings that β_{IV} -spectrin is a novel regulator of angiogenesis and tip cell dynamics.

4. Did the authors perform tamoxifen injection in pups to avoid embryonic lethality? It would be interesting to assess angiogenic responses in adult mice with a pathological condition to determine the therapeutic potential of this newly-identified endothelial target.

Yes, we injected tamoxifen at P1 and P3 throughout our studies. We also tried administering tamoxifen at earlier stages (~E12.5 and E15.5) but observed even higher frequency of embryonic lethality in KO but not WT. Studying β_{IV} -spectrin in an adult pathological context is ongoing as originally mentioned in the Discussion (i.e. vascular aging, atherosclerosis, and tumor angiogenesis) as separate follow-up papers.

5. In the Figure 3, comparison of total VEGFR2 and activated receptor levels in WT and β_{IV} -shRNA-treated MEECs, the authors propose a negative feedback response was responsible for the decrease in receptor expression. Characterization of this response should include additional supporting experimentation to show a time course of MG132 treatment. At the same time, the fluorescence imaging of the retinal vasculature could be improved. We now provide a time course experiment complete with densitometry quantification (Fig.3F). We also improved the images of the retinal vasculature.

6. A more thorough examination of VEGFR2 location in WT and β_{IV} -shRNA cells (i.e., cell surface versus endocytic compartments, recycling endosomes, lysosomes) and turnover would have added to the study.

We provide new confocal Z-stack images of β_{IV} -spectrin, VEGFR2 and recycling marker EEA1 (Fig.4A, Fig.S5). Together, our data show that endogenous β_{IV} -spectrin localizes largely in basolateral membrane domains. Similarly, as previously reported, VEGFR2 (as reported Hudson et al., *Dev Cell*, 2014) also accumulates significantly in basolateral compartments although some endocytic localization is observed at steady-state. Importantly, we observe that in β_{IV} -shRNA cells, VEGFR2's overall expression and its basolateral accumulation is significantly greater, thus supporting the role of β_{IV} -spectrin in promoting its internalization and turnover.

7. The tissue source of the mouse embryonic endothelial cells (MEECs) used for Figure 3 A to F is a mixture of heart, lung, liver and kidney cells purified using CD31-conjugated beads, which wouldn't account for tissue-specific differences or distinguish between arterial, capillary, and venous cells.

We note that MEECs are an immortalized mouse embryonic endothelial cell line which we and others have previously characterized (Pece-Barbara et al., *JBC*, 2005, Lee et al, *MBoC*, 2012).

8. What domains are associated with the N-terminal and C-terminal expression constructs? Finer mapping of the potential VEGFR2 regulatory domains would make for a more complete mechanistic study. While the authors point toward a previous study showing CaMKII interaction with the C terminus of β_{IV} -spectrin, a more precise mapping of the interacting domain is warranted.

Previous studies have mapped the C-terminal segment (2292-2308) of β_{IV} -spectrin as the binding site for CaMKII in both in vitro pull-down and in vivo settings (Hund et al., *JCI*, 2010; Unudurthi et al. *JCI*, 2018). Our findings in endothelial cells are consistent with the previous findings as both knockdown and C-terminal truncations result in the loss of the β_{IV} -spectrin/CaMKII interactions. We have modified our original figure to represent the CaMKII interaction site (Fig.4B).

9. KN-93 has a number of targets other than CaMKII and does not block catalytic activity. As a result, the experiments shown in Figures 4 and S4 would have been more convincing if another means to inhibit activity of this kinase; however, the IP studies somewhat alleviates this concern.

While we acknowledge the limitations of small-molecule inhibitors such as KN-93, given the overarching scope of our findings derived from in vitro, in vivo, cellular and biophysical MS-based experiments, we strongly believe that CaMKII is a key modulator of VEGFR2 turnover.

10. The fluorescent co-localization of β_{IV} -spectrin with CaMKII and VEGFR2 is difficult to distinguish at the magnification and resolution presented in Figure 4G.

We now provide higher quality confocal images for the original figure, Fig.4G, which is now Fig.4F. Due to other reviewers' concerns, we also provide additional fluorescent images including 3-D Z-stacks (Fig.4A, Fig.S5).

11. In Figure 6A, the S1235 site is not indicated in the schematic diagram. The information presented in Figure 6C is difficult to interpret based on so few cells. The graph of this data seems to be underpowered from a statistical standpoint. Analysis of 25 cells per group is likely inadequate.

Fig.6A has been edited accordingly. The data in question for Fig.6C is actually a culmination of three independent experiments (n=3), each experiment involving the measurement of at least 25 cells per group (total of >75 cells). This is now clearly stated in the figure legend.

12. The authors rely on presenting what appears to be single experiments for western analyses (e.g., Figure 7A), which should include appropriate statistical analyses of independent experiments and graphical representation.

We now provide as new data densitometry quantifications for Fig.7A as well as other figures (e.g. Fig.1A, Fig.3F, Fig.4C,D).

Reviewer #4 (Remarks to the Author):

This manuscript presents important insights into stalk cell specification by the protein β IV spectrin and provides sound experimental evidence of the mechanism by tracking the phosphorylation dependent association of VEGFR with the β IV spectrin and CaMKII complex and spatial marking of β IV spectrin expression. The overall role of β IV spectrin in tip-stalk cell dynamics is clearly presented and opens doors to many follow-up studies exploring VEGFR2 regulation. My specific comments are as follows:

1. Define/write in full 'VEGF' in line 69 when you first introduce the term.

This has been added.

2. Define/write in full 'PKC' in line 84 when you first introduce the term.

This has been added.

3. Figure S1A- Clearly explain and mark what the top and bottom panels represent. This is the very first figure being discussed, and it's confusing. Is it confluent v/s proliferative or is it neuronal v/s vascular? The caption and text say different things here.

It is comparing the proliferative vs confluent states. We have now modified Fig.S1A to make it more intuitive and edited the text to clarify the experimental objectives and findings.

4. Line 127- mention the purpose of adding morpholinos and what it is.

The zebrafish data was deemed weak and problematic by the other reviewers and has since been omitted. However, given the robustness of the mouse data, we believe our central findings are valid.

5. Figure 2A- Please write the time points in the figure itself.

This has been added.

6. Line 146,147- If it is still about β IV-ECKO embryos, don't start a new sentence as if it is another experiment. Is it just a different time point or a different embryo?

These sentences have been restructured for clarity and accuracy.

7. Figure 2F- caption should state that the depletion is induced by tamoxifen.

This has been added.

8. Figure 3A is not readable at all, please increase the size.

This has been modified accordingly.

9. Figure S3- what are the 6 fractions shown in the Excel sheet (column I)? Are those the gel pieces digested and analyzed separately? If so, why does the method section say that the gel was cut into 5 pieces?

We apologize for the confusion. 6 fractions represent 6 individual cell lysate samples derived from n=3 for control WT and n=3 for β IV-shRNA. Although each fraction lane was resolved on SDS-PAGE, each of these lanes was cut into 5 gel pieces to better resolve the mass differences and their quantity in our MS quantitative proteomics analysis. We have added raw data image to better illustrate this point in Fig.S3.

10. In-gel digestion method- The references quoted do not include reduction and alkylation steps prior to trypsin digestion. Is there a reason those were not performed?

We performed with and without reduction/alkylation tests and decided previously that the data was good without these additional steps. As a result, we phased out reduction and alkylation from our in-gel based sample prep. This move resulted in a streamlined protocol, which was extremely welcome since we desalt everything offline. We have found that our offline desalting results in peptide purity that serves as an excellent match to our chromatographic strategy, both of which when combined, work very well during the chromatographic alignment step in the Progenesis data processing pipeline.

11. Line 173- "Instead, β IV spectrin appeared..." this sentence referring to Figure 3F is confusing to the reader, and the caption is also absent in the Figure. Please rephrase the sentence and add a caption.

We apologize for this confusion. The text has been edited for clarity.

12. Figure S5- Please provide methods for phosphopeptide analysis. Was any enrichment performed to concentrate phosphopeptides?

Yes, there was an enrichment step, and a more detailed description is cited in the Methods section.

Reviewers' Comments:

Reviewer #1:

Remarks to the Author:

Unfortunately, the concerns that I had with the first submission still remain even after the resubmission of the manuscript.

1) Although the authors have alleviated my reservations about the zebrafish data by removing all zebrafish data altogether, I still question their mouse retina data, particularly the immunostaining of betaIV-spectrin and VEGFR2, which still look unspecific i.e. non-endothelial.

- Although, the authors have provided high magnification/resolution confocal Z-stack images of endogenous betaIV-spectrin in ECs in culture, they did not provide such images for ECs in the retina. Can the authors show higher magnification images and orthogonal views of betaIV-spectrin staining together with an EC marker such as IB4. I will be convinced that the betaIV-spectrin staining is specific to ECs if it is absent from the lumen i.e. it colocalizes with IB4.

- It is clear that the biochemical assays and MS-proteomics show that there is an increase in VEGFR2 levels in betaIV-spectrin KO ECs. However, I am still unconvinced that the increase in VEGFR2 level is stalk cell-specific because I am unconvinced by the VEGFR2 staining. Can you please convince me by showing me higher magnification images and orthogonal views of VEGFR2 staining together with an EC marker such as IB4. As with the betaIV-spectrin staining, I will be convinced that the VEGFR2 staining is specific to ECs if it is absent from the lumen.

Furthermore, if there is indeed an increase in VEGFR2 signaling in stalk cells after betaIV-spectrin depletion in ECs, wouldn't these stalk cells have increased migratory potential and therefore acquire tip cell position? However, in Fig. 3I/J (is labelling of this figure correct?), "VEGFR2-enriched ECs" are distinctly behind tip cells. Why is this the case?

2) As a response to my suggestion, the authors have performed ESM1 staining in wildtype and betaIV-spectrin-ECKO retinas (Fig. S9). While I am pleased that a more specific tip cell marker is used, I would like to ask why there is a difference in ESM1 staining intensity between the WT and betaIV-spectrin-EC KO retinas (that is weaker in WT/brighter in betaIV-spectrin-EC KO). Also, can you please show ESM1 staining together with a pan-EC marker such as IB4 in WT and betaIV-spectrin-ECKO retinas to convincingly show an increase in tip cells? Furthermore, in your rebuttal, it is written that "there are greater number of sprouting tip cells in betaIV-spectrin-ECKO retinas" but there are no numbers to support this claim; can you please provide the quantification?

3) The authors are adamant that ERG is a marker for EC proliferation in vivo. However, I would like to correct this assumption. ERG is used as a marker for EC nuclei and in the papers cited (Birdsey et al., Dev Cell, 2015; Pontes-Quero et al., Nat Commun, 2019), ERG was not used as a marker of EC proliferation but as a marker for EC nuclei. The authors in Pontes-Quero et al. write: "Anti-Erg (red) labels EC nuclei."

Based on the ERG staining that the authors have shown, the in vivo data only shows an increase in EC density in the P5 betaIV-EC KO retina. Evidence for increased EC proliferation in the retinal vasculature is absent. Therefore, the authors should either re-word lines 148 – 149 to "..., SUGGESTING that betaIV -spectrin deficiency MAY promote EC proliferation" or perform an in vivo proliferation assay to actually provide experimental evidence for increased EC proliferation in vivo. Typically, EdU or BrdU assay is performed to examine alterations in proliferation. Yes, EdU/BrdU may also stain other cell types but co-staining with ERG would allow you to detect EdU-positive ECs i.e. EdU+/BrdU+ ECs will indicate proliferated ECs. Numerous papers have published such an experiment to examine proliferation, including Pontes-Quero et al. (e.g. Fig. 2) that the authors have cited. I would like to refer you to another recent publication by Andrade et al., 2021, Nat. Cell Biol, particularly Figures 3F and 7G. The rate of proliferating ECs is calculated as EdU/ERG double-positive cells normalized to the number of ERG-positive cells (see Methods).

The authors should note that an increase in EC density can result from an increase EC proliferation OR

increased in EC migration/rearrangement from the angiogenic sprouting front to the vascular plexus as evidenced by lineage tracing experiments of tip cells (Xu et al., 2014, Nat. Commun.; Pitulescu et al., 2017, Nat. Cell Biol.; Park et al., 2021, Circulation). Given that the authors claim that knockdown of betaIV -spectrin in ECs in culture lead to increased migration (Fig. 1C) and proliferation (Fig. 1D), the increase in EC density in betaIV-EC KO retina can be a result of either increased in proliferation or cell rearrangement, neither of which the authors have demonstrated. Therefore, there is no evidence demonstrating that there is EC hyperproliferation in betaIV-EC KO mice.

Other comments on new data:

Fig. S7 - The images are of poor quality to the extent that a difference cannot be made between WT and 4J mice. Furthermore, I do not understand the quantification for number of branches per P3 Retina. How can there be around 3 branches per WT retina? And there are no error bars; was only 1 retina per genotype quantified?

Reviewer #2:

Remarks to the Author:

The authors have addressed all my substantive concerns and the manuscript is much improved, and their interpretations and conclusions strengthened. However, there remain a few issues in the presentation of the data that should be corrected to improve communication of the results, as well as to ensure transparency of the data.

1. The rigor of the authors' conclusions has been greatly improved by the addition of the quantitative data. However, the bar graphs should be presented as dot plots, with mean and SD. This allows reader to see whether data points are clustered around the mean, or widely scattered, or even in two sub-populations. This is the current standard of presentation for data transparency.

2. The fluorescent images with only one color should be shown in gray scale (e.g., Figure 1C-E, Figure 2H, Figure 3F, Figure 4A and C, Figure 5C, Figure 6B-C, D-G; Suppl Figs S2, S5, S7). Red on black, or green on a black, or blue on a black background are very poor contrast and make the details difficult to see. This will greatly improve the main points the authors make, and improve the visualization of the dim staining in the cells. In the case of images with colocalization, the individual channels can be presented in gray scale and the merge in color; the letters used to state the probe on the figure can be indicated in the appropriate color. Note that for the merged image in Figure 4A with green and blue, the green can be converted to gray scale and blue left as is; blue contrasts extremely well with white and will be easy to see. For Figure 6B-C, if the blue is merged with gray scale it will be a lot easier to see.

I know this is a bit of work, but it is a shame when the colors in the confocal images are such as to render the details hard to discern. Additionally, there are readers who may be red/green color blind or low vision.

Reviewer #3:

Remarks to the Author:

The authors have been responsive to the initial round of review by extensively modifying the manuscript and figures. They also included new data to address some of the earlier concerns of the reviewers and deleted the Zebrafish screening experiments, which were problematic in terms of appropriate controls and data quality. While there remains some unanswered questions regarding endogenous spectrin location and the specificity of the inhibitor, the authors have improved many of the microscopy images and added other results. The figure legends are more complete and text has

been inserted to make the manuscript more understandable to the reader. As a consequence, I believe that this submission is now of sufficient quality and scientific interest to warrant publication in Nature Communications provided they satisfy the concerns of the other reviewers.

Reviewer #4:

Remarks to the Author:

Most of my concerns have been addressed by the authors. However, there are still some that need attention:

1. Figure 2A does not have time points added in the figure itself.
2. Requested information has not been added to Figure 2F.
3. Please highlight the restructured sentences as mentioned originally in point 6 (I can not find the lines after revision and renumbering of lines).
4. In gel digestion- authors claim that their data is good enough without the reduction/alkylation steps. I am curious about the total number of proteins confidentially identified in the study. Differentially regulated proteins are in hundreds as stated, please report the total # proteins. That is an indication of how well the digestion worked.
5. The authors state that a detailed description for phosphopeptide enrichment is cited in the Methods section. I do not see it. Neither the description nor the citations have any mention of phosphopeptide enrichment steps.

Reviewer #1 (Remarks to the Author):

Unfortunately, the concerns that I had with the first submission still remain even after the resubmission of the manuscript.

1) Although the authors have alleviated my reservations about the zebrafish data by removing all zebrafish data altogether, I still question their mouse retina data, particularly the immunostaining of betaIV-spectrin and VEGFR2, which still look unspecific i.e. non-endothelial.

- Although, the authors have provided high magnification/resolution confocal Z-stack images of endogenous betaIV-spectrin in ECs in culture, they did not provide such images for ECs in the retina. Can the authors show higher magnification images and orthogonal views of betaIV-spectrin staining together with an EC marker such as IB4. I will be convinced that the betaIV-spectrin staining is specific to ECs if it is absent from the lumen i.e. it colocalizes with IB4.

We now provide higher magnification confocal Z-stack images of BIV-spectrin and IB4 as part of new data in Figure 2G. As the reviewer will note, the fluorescent staining of BIV-spectrin and IB4 are observed along the circumference of the vessel cross section but are largely absent in the lumen. White arrows indicate the regions along the circumference where BIV-spectrin and IB4 co-localize (yellow). **Please note that another reviewer requested that most of the single-color fluorescence images be changed to gray scale for red/green color-blind and low vision reviewers and readers.**

- It is clear that the biochemical assays and MS-proteomics show that there is an increase in VEGFR2 levels in betaIV-spectrin KO ECs. However, I am still unconvinced that the increase in VEGFR2 level is stalk cell-specific because I am unconvinced by the VEGFR2 staining. Can you please convince me by showing me higher magnification images and orthogonal views of VEGFR2 staining together with an EC marker such as IB4. As with the betaIV-spectrin staining, I will be convinced that the VEGFR2 staining is specific to ECs if it is absent from the lumen.

We now provide as new data in Figure 3J higher magnification confocal Z-stack images of VEGFR2 and IB4. Again, their fluorescent stainings are visible along the wall of the vessel cross section but largely absent in the lumen.

Furthermore, if there is indeed an increase in VEGFR2 signaling in stalk cells after betaIV-spectrin depletion in ECs, wouldn't these stalk cells have increased migratory potential and therefore acquire tip cell position? However, in Fig. 3I/J (is labelling of this figure correct?), "VEGFR2-enriched ECs" are distinctly behind tip cells. Why is this the case?

We have replaced the original image in Figure 3I with a new one that better exemplifies the extensive increase in VEGFR2 along the leading edge, with white arrows indicating ECs with tip cell features (i.e. filopodial projections).

2) As a response to my suggestion, the authors have performed ESM1 staining in wildtype and betaIV-spectrin-ECKO retinas (Fig. S9). While I am pleased that a more specific tip cell marker is used, I would like to ask why there is a difference in ESM1

staining intensity between the WT and betaIV-spectrin-EC KO retinas (that is weaker in WT/brighter in betaIV-spectrin-EC KO). We believe the elevated ESM1 fluorescence intensity of the EC KO retinas are due to an overall increase in ESM1-positive cell count AND protein expression. This is evidenced in our data where both ESM1+ cell count and corrected fluorescence intensity are increased when betaIVspectrin is depleted relative to control (Figure S9C).

Also, can you please show ESM1 staining together with a pan-EC marker such as IB4 in WT and betaIV-spectrin-ECKO retinas to convincingly show an increase in tip cells? Unfortunately, we have been unsuccessful at dual-staining for ESM1 and IB4, in part because, in our hands, IB4 staining works only when retinas are briefly fixed with paraformaldehyde prior to methanol fixation overnight whereas, for whatever reason, ESM1 staining is not compatible with methanol fixation at all.

Furthermore, in your rebuttal, it is written that “there are greater number of sprouting tip cells in betaIV-spectrin-ECKO retinas” but there are no numbers to support this claim; can you please provide the quantification? We now include this graph quantification as new data in Figure S9B.

3) The authors are adamant that ERG is a marker for EC proliferation in vivo. However, I would like to correct this assumption. ERG is used as a marker for EC nuclei and in the papers cited (Birdsey et al., Dev Cell, 2015; Pontes-Quero et al., Nat Commun, 2019), ERG was not used as a marker of EC proliferation but as a marker for EC nuclei. The authors in Pontes-Quero et al. write: “Anti-Erg (red) labels EC nuclei.” Based on the ERG staining that the authors have shown, the in vivo data only shows an increase in EC density in the P5 betaIV-EC KO retina. Evidence for increased EC proliferation in the retinal vasculature is absent. Therefore, the authors should either reword lines 148 – 149 to “..., SUGGESTING that betaIV -spectrin deficiency MAY promote EC proliferation” or perform an in vivo proliferation assay to actually provide experimental evidence for increased EC proliferation in vivo. Typically, EdU or BrdU assay is performed to examine alterations in proliferation. Yes, EdU/BrdU may also stain other cell types but co-staining with ERG would allow you to detect EdU-positive ECs i.e. EdU+/BrdU+ ECs will indicate proliferated ECs. Numerous papers have published such an experiment to examine proliferation, including Pontes-Quero et al. (e.g. Fig. 2) that the authors have cited. I would like to refer you to another recent publication by Andrade et al., 2021, Nat. Cell Biol, particularly Figures 3F and 7G. The rate of proliferating ECs is calculated as EdU/ERG double-positive cells normalized to the number of ERG-positive cells (see Methods).

The authors should note that an increase in EC density can result from an increase EC proliferation OR increased in EC migration/rearrangement from the angiogenic sprouting front to the vascular plexus as evidenced by lineage tracing experiments of tip cells (Xu et al., 2014, Nat. Commun.; Pitulescu et al., 2017, Nat. Cell Biol.; Park et al., 2021, Circulation). Given that the authors claim that knockdown of betaIV -spectrin in ECs in culture lead to increased migration (Fig. 1C) and proliferation (Fig. 1D), the

increase in EC density in betaIV-EC KO retina can be a result of either increased proliferation or cell rearrangement, neither of which the authors have demonstrated.

Therefore, there is no evidence demonstrating that there is EC hyperproliferation in betaIV-EC KO mice.

We thank the reviewer for the insights and the kind suggestion that we can modify the sentence to “suggesting that betaIV-spectrin deficiency may promote EC proliferation.” This edited line is now underlined in the text.

Other comments on new data:

Fig. S7 - The images are of poor quality to the extent that a difference cannot be made between WT and 4J mice. Furthermore, I do not understand the quantification for number of branches per P3 Retina. How can there be around 3 branches per WT retina? And there are no error bars; was only 1 retina per genotype quantified?

We apologize for the labeling error- it was supposed to be of overall fluorescence (CTCF). We replaced the original data with a dot-plot graph, which shows n=4 for WT and n=3 for 4J retinas.

Reviewer #2 (Remarks to the Author):

The authors have addressed all my substantive concerns and the manuscript is much improved, and their interpretations and conclusions strengthened. However, there remain a few issues in the presentation of the data that should be corrected to improve communication of the results, as well as to ensure transparency of the data.

1. The rigor of the authors' conclusions has been greatly improved by the addition of the quantitative data. However, the bar graphs should be presented as dot plots, with mean and SD. This allows reader to see whether data points are clustered around the mean, or widely scattered, or even in two sub-populations. This is the current standard of presentation for data transparency.

All bar graphs have now been replotted as dot plots with gray scaled bars.

2. The fluorescent images with only one color should be shown in gray scale (e.g., Figure 1C-E, Figure 2H, Figure 3F, Figure 4A and C, Figure 5C, Figure 6B-C, D-G; Suppl Figs S2, S5, S7). Red on black, or green on a black, or blue on a black background are very poor contrast and make the details difficult to see. This will greatly improve the main points the authors make, and improve the visualization of the dim staining in the cells. In the case of images with colocalization, the individual channels can be presented in gray scale and the merge in color; the letters used to state the probe on the figure can be indicated in the appropriate color. Note that for the merged image in Figure 4A with green and blue, the green can be converted to gray scale and blue left as is; blue contrasts extremely well with white and will be easy to see. For

Figure 6B-C, if the blue is merged with gray scale it will be a lot easier to see.

I know this is a bit of work, but it is a shame when the colors in the confocal images are such as to render the details hard to discern. Additionally, there are readers who may be red/green color blind or low vision.

We converted most of the fluorescent images into gray scale. On occasion, we had to keep the original colors in order to best demonstrate the changes/differences that another reviewer requested.

Reviewer #3 (Remarks to the Author):

The authors have been responsive to the initial round of review by extensively modifying the manuscript and figures. They also included new data to address some of the earlier concerns of the reviewers and deleted the Zebrafish screening experiments, which were problematic in terms of appropriate controls and data quality. While there remains some unanswered questions regarding endogenous spectrin location and the specificity of the inhibitor, the authors have improved many of the microscopy images and added other results. The figure legends are more complete and text has been inserted to make the manuscript more understandable to the reader. As a consequence, I believe that this submission is now of sufficient quality and scientific interest to warrant publication in Nature Communications provided they satisfy the concerns of the other reviewers.

Reviewer #4 (Remarks to the Author):

Most of my concerns have been addressed by the authors. However, there are still some that need attention:

1. Figure 2A does not have time points added in the figure itself.

Time point labels have now been added.

2. Requested information has not been added to Figure 2F.

The phrase "Tamoxifen-induced depletion" has been added.

3. Please highlight the restructured sentences as mentioned originally in point 6 (I can not find the lines after revision and renumbering of lines).

The restructured sentence is now underlined and highlighted in yellow.

4. In gel digestion- authors claim that their data is good enough without the reduction/alkylation steps. I am curious about the total number of proteins confidentially identified in the study. Differentially regulated proteins are in hundreds as stated, please report the total # proteins. That is an indication of how well the digestion worked.

Of the over 7000 proteins identified in the study, 5648 were identified with having at least 2 unique peptides. Peptide identifications were accepted if they could be established at greater than 95.0% probability. Protein identifications (proteins containing at least 2 peptides) were accepted if they could be established at greater than 99.0% probability.

5. The authors state that a detailed description for phosphopeptide enrichment is cited in the Methods section. I do not see it. Neither the description nor the citations have any mention of phosphopeptide enrichment steps.

We apologize for this oversight. The Methods section now contains the description: To determine global differences in protein phosphorylation abundance between WT and bIV-spectrin depleted cells, 100 mg of protein lysate per sample (n=3) was subjected to in-solution tryptic digestion and phosphopeptide enrichment using sequential enrichment from metal oxide affinity chromatography per manufacturer's protocol (Thermo Scientific, San Jose, CA).

Reviewers' Comments:

Reviewer #1:

Remarks to the Author:

The authors have almost addressed all my questions. However, I still have some concerns:

1) It is surprising to hear that in your hands, IB4 staining only works after methanol fixation. In fact, many have successfully performed IB4 staining without methanol staining (with only 2 or 4% PFA fixation). Please refer to e.g. Gerhardt et al., 2003, J. Cell Biol. and Carvalho et al., 2019 eLife for a fixation protocol without methanol for immunofluorescent staining of wholemount retina. With this protocol, I believe that you will be able to perform co-immunostaining with other antibodies including that against ESM1 to observe co-localization of proteins.

2) The authors have included the quantification of sprouting tip cells in WT and betaIV-spectrin-EC KO retinas (Fig. S9B). However, it is unclear to me how the authors can determine ESM+ endothelial cell number in the retinas from Fig. S9 since there is no EC nuclear marker to indicate cell number. Can the authors provide an explanation on the method used to quantify ESM+ cells?

Reviewer #2:

Remarks to the Author:

My concerns have all been addressed. I also note that the additional high mag images showing that the betaIV spectrin is endothelial and not luminal (as requested by one of the other reviewers) adds greatly to the study.

One suggestion for the authors is to brighten the gray scale images of the retinas in Figures 2C-E, Figure 3H-I, Figure 4A, and Figure 5C. They are very dim and increasing the brightness (equivalently for control and experimental samples) will greatly improve the communication of the points they want to convey.

Reviewer #4:

Remarks to the Author:

My concerns have mostly been addressed by the authors. Information added regarding phosphopeptide enrichment protocol helps but needs a bit more information. Manufacturer's protocol has no weblink or citation (simply writing the company's name doesn't help the reader). Please add a citation that uses the protocol or provide a link to the webpage in references. Also add the Thermo products used in the key Resources table. The differentially regulated proteins are an important piece of evidence in the manuscript, the protocol therefore should be given due attention.

REVIEWERS' COMMENTS

Reviewer #1 (Remarks to the Author):

The authors have almost addressed all my questions. However, I still have some concerns:

1) It is surprising to hear that in your hands, IB4 staining only works after methanol fixation. In fact, many have successfully performed IB4 staining without methanol staining (with only 2 or 4% PFA fixation). Please refer to e.g. Gerhardt et al., 2003, J. Cell Biol. and Carvalho et al., 2019 eLife for a fixation protocol without methanol for immunofluorescent staining of wholemount retina. With this protocol, I believe that you will be able to perform co-immunostaining with other antibodies including that against ESM1 to observe co-localization of proteins.

We thank the reviewer for this advice and will follow the described protocol in future experiments.

2) The authors have included the quantification of sprouting tip cells in WT and betaIV-spectrin-EC KO retinas (Fig. S9B). However, it is unclear to me how the authors can determine ESM+ endothelial cell number in the retinas from Fig. S9 since there is no EC nuclear marker to indicate cell number. Can the authors provide an explanation on the method used to quantify ESM+ cells?

We note that a tri-staining was not possible in this instance due to species cross-reactivity between the three antibodies. Therefore, we scored ESM+ stained cells displaying prominent filopodial projections as a hallmark feature of tip cells in our quantifications. Moreover, we were conservative in the scoring process in that if it was unclear whether the ESM1+ staining comprised one or two ECs that displayed the jagged, filopodial projections, we scored them as a single tip cell.

Reviewer #2 (Remarks to the Author):

My concerns have all been addressed. I also note that the additional high mag images showing that the betaIV spectrin is endothelial and not luminal (as requested by one of the other reviewers) adds greatly to the study.

One suggestion for the authors is to brighten the gray scale images of the retinas in Figures 2C-E, Figure 3H-I, Figure 4A, and Figure 5C. They are very dim and increasing the brightness (equivalently for control and experimental samples) will greatly improve the communication of the points they want to convey.

We have brightened the gray scale images of the retinas equivalently for control and experimental samples.

Reviewer #4 (Remarks to the Author):

My concerns have mostly been addressed by the authors. Information added regarding phosphopeptide enrichment protocol helps but needs a bit more information. Manufacturer's protocol has no weblink or citation (simply writing the company's name doesn't help the reader). Please add a citation that uses the protocol or provide a link to the webpage in references. Also add the Thermo products used in the key Resources table. The differentially regulated proteins are an important piece of evidence in the manuscript, the protocol therefore should be given due attention.

We apologize for the omission of the details of the protocol. Here is the link:
<http://list.abrf.org/groups/abrf/files/f/mAAPvXXqNnTAVu3LHkdjQsL7BnV-23hX-2FMxhrl/SMOAC%20-%20Sequential%20enrichment%20of%20Metal%20Oxide%20%20Affinity%20Chromatography.pdf>

The link has been added in the references and highlighted in yellow. Also, the products have been added in the key Resources table and highlighted in yellow.